# Assessment of undergraduate student knowledge, attitude, and practices towards COVID-19 in Debre Berhan University, Ethiopia

**Yared Asmare Aynalem**[1]*, **Tadess Yirga Akalu**[2], **Birhan Gebresellassie Gebregiorgis**[1], **Nigussie Tadesse Sharew**[1], **Hilina Ketema Assefa**[1], **Wondimeneh Shibabaw Shiferaw**[1]

1 Department of Nursing, College of Health Science, Debre Berhan University, Debre Berhan, Ethiopia,
2 Department of Nursing, College of Health Science, Debre Markos University, Debre Markos, Ethiopia

* yaredasmare123@gmail.com

## Abstract

### Background

Novel coronavirus-2019 (COVID-19) is a highly infectious disease that has caused a global pandemic. As of July 2020, there were 8,475 confirmed cases of COVID-19 in Ethiopia, and a total of 52 cases and 1 death were reported in Debre Berhan where this study was conducted. Under these conditions, we sought to assess what undergraduate students at Debre Berhan University knew about COVID-19 and how it shaped their attitudes and practices regarding this disease.

### Objective

The aim of the current study was to assess undergraduate student knowledge, attitudes and practices towards COVID-19 in Debre Berhan University, Ethiopia.

### Methods

A cross-sectional survey was conducted from March 18–24, 2020 among undergraduate students at Debre Berhan University. A two-stage cluster sampling technique was employed with a total sample size of 634. Proportional allocation of samples was used to the randomly selected colleges, and a systematic random sampling technique was employed to recruit the students. The data were checked for completeness, coded, entered into Epi-Data VS 3.1, and then exported into STATA™ Version 14 software for analysis. Descriptive statistics were conducted. Binary logistic regression analyses were used to identify factors. Factors were selected with the entry method. Adjusted odds ratios (AORs) and their 95% confidence intervals (CIs) were used to assess the associations between variables and knowledge, attitude and practices (KAP).

**Data Availability Statement:** All relevant data are within the paper and supporting information files

**Funding:** The author(s) received no specific funding for this work.

**Competing interests:** The authors have declared that no competing interests exist.

**Abbreviations:** AORs, Adjusted odds ratios; CIs, confidence intervals; DBU, Debre Berhan University; KAP, Knowledge, Practice and Attitude; WHO, World Health Organization.

## Results

From a total of 546 included participants, more than half of them, 307 (57%) were males. Seventy-three percent of them heard about novel coronavirus from social media. In this study, 73.8% of the participants were knowledgeable, and their overall attitude was favorable. Approximately 71.4% correctly responded that the main clinical symptoms of COVID-19 are fever, fatigue, dry cough, and shortness of breath. Nearly half, 229 (42%) of the students approved that they had no concern of being infected with COVID-19. Moreover, most participants showed poor practices; more than half of the study subjects were not maintaining a physical distance. In multivariable analyses, people older than 25 years (AOR = 1.6, 95% CI; 1.2, 4.6) and those who lived in urban areas (AOR = 4.3, 95% CI; 2.6, 15.8) were significantly more knowledgeable about COVID-19. Furthermore, those students that have information about COVID-19 (AOR = 2.3, 95% CI; 1.6, 8.7) was significantly associated with the attitude undergraduate students had about COVID-19 as compared with its counterpart.

## Conclusion

The undergraduate students at Debre Berhan University were moderately knowledgeable about COVID-19 and had an optimistic attitude towards its resolution. However, this optimism may lead to poor public health practices within this community. Therefore, greater efforts need to be made through more comprehensive and directed actions and awareness campaigns to increase the knowledge, attitude and practice of the students.

## Background

The novel coronavirus-2019 (COVID-19) is a newly discovered infectious disease that can cause severe illness in humans [1]. It was first detected in December 2019 in Wuhan, China, and quickly spread around the world. Fever, dry cough, fatigue, myalgia, and dyspnea are considered the main clinical symptoms of this disease. In China, 18.5% of the patients with COVID-19 developed acute respiratory distress syndrome, septic shock, metabolic acidosis, and bleeding. In more severe cases, it can cause pneumonia, severe acute respiratory syndrome, kidney failure, and even death [2, 3].

As of July 17, 2020, there were 13,961,427 confirmed cases and 592,979 deaths (7%) globally. This death rate is particularly alarming, as estimated at more than 7% [4]. The World Health Organization (WHO) has declared as the sixth public health emergency of international concern since the beginning of this century, following H1N1 (2009) [5], polio (2014), Ebola in West Africa (2014), Zika (2016), and Ebola in the Democratic Republic of Congo (2019) [6]. According to the World Health Organization (WHO) report, nearly forty million cases of what have been reported in more than two hundred fifteen countries worldwide, and the number of infected cases has increased gradually [7, 8].

Given the severity of COVID-19 and its fast spreading, the WHO has called for collaborative efforts within and between countries to prevent the rapid spread of COVID-19 [9]. The WHO set the following strategic objectives for battling this disease: Limited human-to-human transmission prevents disease transmission from high-burden countries by limiting travel, identifying, isolating, and caring for patients, reducing the transmission from animal sources, developing diagnostic tools, therapeutics, and vaccines, communicating risks and events to all

communities, and minimizing the social and economic impact of COVID-19 through multi-sector partnerships [2, 10]. With respect to individuals, standard recommendations to stop the spread of COVID-19 include regular hand washing, covering the mouth and nose when coughing and sneezing, and physical distancing [11].

An international study revealed that knowledge regarding COVID-19 was adequate among most students, and the vast majority of the participants also held an optimistic attitude towards the COVID-19 [12–15]. However, findings from Saudi Arabia [16], Middle Eastern Arabic countries (Egypt Iran and Jordan) [17], India [18], Tanzania [19] and Uganda [20] showed that most of the community had inadequate knowledge and unfavorable attitudes towards COVId-19. Different findings also revealed that to achieve the desired control of COVID-19, adherence to control measures was essential, which is largely affected by people's knowledge, attitudes, and practices (KAP) towards the disease [13, 15, 21].

Ethiopia has applied preventive measures of COVID-19 using physical distancing, closing schools, home stay encouraging had washing practice, masks and launch of state emergency at the national level earlier to prevent and control the diseases. Despite this effort, the occurrence of COVID-19 is increasing. Notably, in Ethiopia, as of July 17, 2020, there were a total of 8,475 cases and 148 deaths reported by the Ministry of Health and the Ethiopian Public Health Institute. Moreover, as of July 17, 2020, there are a total of 52 cases and 1 death reported in Debre Berhan, which is the study area for this article. The pandemic also puts the whole educational system in to difficult situations; predominantly, undergraduate students represented a special group that was at the ages to get independence and freedom of life but with inadequate experiences. Consequently, their KAP were suggested to be significantly affected by the pandemic as evidenced from a short assessment survey, which needed to be explored. With this gap, little is known about undergraduate students' KAP. Given this fact, we aimed to assess the KAP of Debre Berhan University (DBU) students towards COVID-19.

## Methods

### Study design, setting, and population

A cross-sectional study was conducted from March 18–24, 2020, with undergraduate DBU students. The university is located in Debre Berhan, which are 130 km from Addis Ababa and 682 km from Bahir Dar of the Amhara regional state. According to the 2019 Registrar Office report, DBU has approximately 29,304 students pursuing regular and extension undergraduate and graduate studies. Of these, more than 11,000 are regular undergraduate students. The university has two institutes, ten Schools, and 53 departments. The source population of this study was all undergraduate students at Debre Berhan University, and the study population was all students in randomly selected Schools (Schools of Engendering, computing science, Faculty of Business and Economics and Schools of Agriculture). A table of relevant demographic details is attached as a **S1 File**.

### Sample size determination and sampling procedure (procurement procedures)

The sample size was calculated using a single population proportion formula assuming a 95% confidence level, a 3% margin of error, and a 50% proportion for the KAP level. Since there is no study in our country, we have used a 50% proportion [22]. After a 10% non-response rate and 1.5 design effects, the final sample size was 634. Our study sample was obtained by a two-stage cluster sampling technique. In the first stage, out of ten Schools, four of them (Schools of Engendering, Computing Science, Business and Economics and Schools of Agriculture) were

selected by simple random sampling, and the sample size was allocated proportionally to obtain the required sample size from each selected Schools. In the second stage, the study participants were selected from each year and sections of the selected Schools using a systematic random sampling technique every 7[th] student from each Schools registrar office log-book. After that, a dorm-to-dorm visit was used to obtain the sampled students. Students in each selected Schools that were present during the data collection period and were able to give responses were included in the study. Those who had mental or physical disabilities, could not fill out the questionnaires and unwilling to participate in to the study) were excluded from the study.

## Study variables

In this study, knowledge (good/poor), attitude (positive/negative), and practice (good/poor) towards COVID-19 were the outcome variables. Socio-demographic factors, such as age, **residence, sex, marital status, educational level, field of study, income, family size, and religion**, and the source of information regarding COVID-19 were considered independent covariates. Knowledge of attitudes, knowledge and attitudes towards the preventive measures of the COVID-19 was also assessed.

## Data collection tools

A data collection tool was developed from previous studies [23, 24] and WHO course material on emerging respiratory viruses, including COVID-19 [25]. The proposed tool was developed and validated by a multidisciplinary working group of infectious disease physicians, lecturers (Public health, nurse & environmental health professionals, and infectious diseases public health professionals. The working group regularly reviewed literature to select important characteristics and outcomes for inclusion. The development phase was conducted at Debre berhan University health science Collage and the pilot validation phase was conducted with nursing department students. We have also done a pretest from 5% of the student. The team members formed a working group that met via telephone or video conferences at least biweekly. The questionnaire consisted of four parts: 1) socio-demographics; 2) knowledge; 3) attitudes; and 4) practices of COVID-19. The second part of the survey assessed student knowledge about COVID-19, which included the symptoms of COVID-19-affected patients, transmission routes, precautions, and risk prevention. Participants were given three options per question: "Yes", "No", and "I don't know". Correct responses were given one point, while incorrect responses or "I do not know" were given zero points [15]. According to this study, "adequate knowledge" regarding COVID-19 means that the participant's score was above the mean score on knowledge questions. Conversely, "inadequate knowledge" was assigned to students who scored below the mean score on knowledge questions. In this study, Cronbach's alpha coefficient for the knowledge questionnaire was 0.83, indicating acceptable internal consistency [26].

Attitudes towards COVID-19 were measured with four questions. These questions used a five-point Likert-type response scale: strongly agree (5 points), agree (4 points), neutral (3 points), disagree (2 points), and strongly disagree (1 point). Using previous studies, as a baseline, we categorized agree (strongly agree" and "agree together) neutral and disagree (both disagree and strongly disagree) [20]. Subscale scores were calculated for each participant. Higher scores indicated a "favorable" attitude about COVID-19. Participant practices were assessed by yes or no questions on five specific behaviors [27] (Table 4). Data were also collected through self-administration guided by six trained data collectors and 546 of the participate completed

the survey. Additionally, there were no confirmed or suspected cases in the study area at the time of the data collection period.

## Data quality assurance

A draft of the questionnaire was distributed to seven randomly selected faculty members to assess its readability and validity before pretesting for clarity, relevance, and acceptability. The questionnaire was initially developed in English and then translated to the local language (Amharic) by an expert and then back to English to ensure consistency. All data collectors were trained on how to properly collect data and how to maintain confidentiality. Furthermore, the principal investigator followed the data collection process on a daily basis to ensure the completeness of the questionnaires and to give further clarification to the questions when needed.

## Data processing and analysis

The collected data were checked for completeness, coded, entered into Epi- Data Version 3.1, and then exported into STATA™ Version 14 software for analysis [28]. Descriptive statistics were used to describe variables (socio-demographic characteristics, participants' knowledge, attitude and practice regarding COVID-19). Binary logistic regression analyses were used to identify factors associated with KAP. The coding of outcome variables in the regression models was stated as follows. One point was given to adequate, and 0 was inadequate in the knowledge; 1 for positive, and 0 for negative in the attitude; 1 for good, and 0 for poor in the practice part. Correlations between independent variables (including the correlation between knowledge, attitude and practice) were assessed, but we did not find any correlations. The Hosmer-Lemeshow model fit-ness test was also fitted. Confounding and effect modification was evaluated by observing the regression coefficient variation greater than or equal to 15%, and multi-collinearity was checked using the variance inflation factor using a value of $< 10$ as a cut point. Variables with a p-value $< 0.2$ during bivariable analysis were used in our multivariable analyses to control for confounding effects. Factors were selected in the model using entry / standard regression. Adjusted odds ratios (AORs) and their 95% confidence intervals (CIs) were used to quantify the associations between variables and KAP. Adjusted odds ratios (AORs) with 95% confidence intervals (CIs) were also used to determine the strength of the association between dependent and independent variables.

## Ethics consideration

Ethical clearance for this study was obtained from the Institute of Medicine and College of Health Sciences, DBU. It was approved by the University ethical approval committee with the approval number of Ref. No. DBU.R.D136/02/12. Participants also gave written informed consent prior to data collection. After that the participants were informed about their participation was completely voluntary. Additionally, their right to withdraw from the survey at any time were also maintained. All information that was collected was kept strictly confidential, and personal details such as phone number and address were not revealed to people outside the research teams.

# Results

## Socio-demographic characteristics

From a total of 634 samples, 546 of them completed the questionnaire with a response rate of 86.1%. The mean age of the participants was 21.7 ± 2.5 (standard deviation (SD)) years old

**Table 1. Socio-demographic characteristics of the study participants.**

| Variables | Category | Frequency | Percentage |
|---|---|---|---|
| Gender | Female | 239 | 43 |
| | Male | 307 | 57 |
| Age group (years) | <20 | 115 | 21 |
| | 20–24 | 394 | 72.1 |
| | ≥25 | 38 | 6.9 |
| Marital status | Single | 512 | 93.7 |
| | Married | 25 | 4.5 |
| | Divorced | 10 | 1.8 |
| Residence | Rural | 330 | 60.4 |
| | Urban | 216 | 39.6 |
| Year of study | First year | 230 | 42 |
| Third year | Second year | 129 | 23.6 |
| | Third year | 136 | 25 |
| | Fourth and fifth years | 51 | 9.4 |
| Monthly pocket money/ Ethiopian Birr | <200 | 167 | 30.5 |
| | 200–400 | 226 | 41.5 |
| | ≥401 | 153 | 28 |
| Source of information | News media | 115 | 21 |
| | Social media | 402 | 73.6 |
| | Official government websites | 11 | 2 |
| | Friends and family | 18 | 3.4 |

**Note:** Social media was Facebook, Twitter, WhatsApp, YouTube, Instagram, or a telegram. News media includes TV, magazines, newspapers, and radio.

with a range of 18–27 years old. More than half (307 (57%)) of the students were males, and nearly half (230 (42%)) of the students were in their first year. The majority of study participants (512 (93.7%)) were single, and 330 (60.4%) of the participants lived in rural areas. Additionally, most of the study participants (402 (73.6%)) heard about COVID-19 from social media (Table 1).

## Participants' knowledge regarding COVID-19

The mean COVID-19 knowledge score was 9.6 ± 1.8 with a range of 0–13. The correct answer rates for the 13 questions on the COVID-19 knowledge questionnaire were 54.4–95%. Most of the participants (403 (73.8%)) scored above the mean and were considered to have good knowledge about COVID-19. Among the 546 participants, 71.4% correctly responded that the main clinical symptoms of COVID-19 are fever, fatigue, dry cough, and shortness of breath, and the majority (95%) said currently there is no cure for COVID-19 (Table 2).

## Participants' attitude towards COVID-19

Nearly half (229 (42%)) of the students had no concern of being infected with COVID-19. Most of the respondents (447 (81.8%)) agreed that COVID-19 will be successfully controlled, and the majority (458 (83.8%)) believed that Ethiopia could control the spread of COVID-19 virus. The attitudes of students towards COVID-19 are summarized in Table 3.

**Table 2. Knowledge of the study participant towards COVID-19.**

| Variable | Yes | | No | |
|---|---|---|---|---|
| | Frequency | % | Frequency | % |
| The main clinical symptoms of COVID-19 are fever, fatigue, dry cough, and shortness of breath | 390 | 71.4 | 156 | 28.6 |
| Unlike the common cold, stuffy nose, runny nose, and sneezing are less common in people infected with the COVID-19. | 398 | 72.8 | 148 | 27.2 |
| Currently, there is no cure for COVID-19, but early symptomatic and supportive treatment can help most patients recover from the infection. | 519 | 95 | 27 | 5 |
| Not all people with COVID-19 will develop a severe form of the disease. Those who are elderly, have chronic illnesses, and are obese are more likely to be severe cases. | 324 | 59.3 | 222 | 40.7 |
| COVID-19 is transmitted through air, contact, and feco-oral routes. | 418 | 76.5 | 128 | 23.5 |
| Eating or coming into contact with wild animals will lead to COVID-19 infection. | 395 | 72.4 | 151 | 27.6 |
| People with COVID-19 cannot infect others when a fever is not present. | 249 | 45.6 | 297 | 54.4 |
| The COVID-19 virus spreads via respiratory droplets from infected individuals. | 350 | 64.1 | 196 | 35.9 |
| Ordinary residents can wear general medical masks to prevent getting infected by the COVID-19 virus. | 332 | 60.8 | 214 | 39.2 |
| It is not necessary for children and young adults to take measures to prevent being infected by the COVID-19 virus. | 227 | 41.5 | 319 | 58.5 |
| To prevent being infected by COVID-19, individuals should avoid going to crowded places and taking public transportation. | 430 | 78.7 | 116 | 21.3 |
| Isolation and treatment of people who are infected with the COVID-19 virus are effective ways to reduce the spread of the virus. | 445 | 81.5 | 101 | 18.5 |
| People who have been in contact with someone infected with COVID-19 should self-isolate. In general, the observation period is 14 days. | 473 | 86.7 | 73 | 13.3 |

## Participants' practices in relation to COVID-19

The majority of the participants (91.4%) had not visited a crowded place in recent days, and approximately 74% were washing their hands after sneezing and coughing. Nearly half (284 (52%)) of the participants reported covering their mouth and nose with an elbow or tissue while coughing or sneezing. However, approximately 56% of people surveyed did not maintain a social distance of at least one meter between themselves and anyone who was coughing or sneezing (Table 4).

## Multivariable regression results of the binary logistic regression analyses

In our multivariable analyses, students who were 25 years old or older were 1.6 times more likely to be knowledgeable about COVID-19 than those <20 years old (AOR = 1.6, 95% CI: 1.2, 4.6). Similarly, participants who lived in urban areas were 4.3 times more knowledgeable about COVID-19 than those who resided in rural areas (AOR = 4.3, 95% CI: 2.6, 15.8). Lastly, participants who received their information on COVID-19 from the news media were 2.3 times more likely to have a favorable attitude towards COVID-19 compared to those who received their information from friends and family (AOR = 2.3, 95% CI: 1.6, 8.7) (Table 5).

**Table 3. Attitudes of the DBU students towards COVID-19.**

| Attitude questions | Agree (%) | Disagree (%) | Not sure (%) |
|---|---|---|---|
| Do you agree that COVID-19 will be successfully controlled? | 447 (81.8) | 66 (12.2) | 33 (6) |
| I have no concern of being infected with COVID-19. | 229 (42) | 303 (55.6) | 14 (2.4) |
| Do you agree that washing hands with soap and water could help to prevent COVID-19 virus transmission? | 415 (76) | 87 (16) | 44 (8) |
| Do you have confidence that Ethiopia can win the battle against the COVID-19 virus? | 458 (83.8) | 61 (11.2) | 27 (5) |

**Table 4. Practices of DBU students with respect to COVID-19.**

| Questions | Yes | | No | |
|---|---|---|---|---|
| | Frequency | % | Frequency | % |
| In recent days, have you gone to any crowded place? | 47 | 8.6 | 499 | 91.4 |
| Do you wash your hands after sneezing or coughing? | 404 | 74 | 142 | 26 |
| Do you touch your face, nose, or mouth with your unclean hands? | 377 | 69 | 169 | 31 |
| Do you cover your mouth and nose with an elbow or tissue while coughing or sneezing? | 284 | 52 | 262 | 48 |
| In recent days, have you maintained a social distance at least one meter (three feet) between yourself and anyone who is coughing or sneezing? | 240 | 44 | 306 | 56 |

# Discussion

The KAP of a population can often determine the severity of an infectious disease outbreak. Indeed, KAP surveys have been used as important sources of data to design health

**Table 5. Factors associated with knowledge and attitudes towards COVID-19 among students at DBU.**

| Variables | Knowledge | | AOR | Attitude | | AOR |
|---|---|---|---|---|---|---|
| | Good | Poor | | Favorable | Not | |
| Gender | | | | | | |
| male | 171(71.5) | 68 (28.5 | 1.3(0.5–3.8) | 157 (65.7) | 82 (34.3) | 1.5 (0.6–2.7) |
| female | 236 (76.8 | 71 (23.2) | 1 | 207 (68.5) | 100(31.5) | 1 |
| Age (years) | | | | | | |
| <20 | 71 (61.8) | 44 (38.2) | 1 | 79 (68.6) | 36 (31.4) | 1 |
| 20–24 | 264 (67) | 130 (33) | 3.8 (0.6–23.3) | 248 (63) | 146 (37) | 2.7 (0.6–13.4) |
| ≥25 | 28 (73.7) | 10 (26.3) | 1.6 (1.2–4.6)* | 27 (70.3) | 11 (29.7) | 1.4 (0.7–6.8) |
| Marital status | | | | | | |
| Single | 336 (71.6) | 176(34.4) | 2.3 (0.8–9.6) | 297 (58) | 215 (42) | 1.2 (0.6–8.8) |
| Married | 16 (64) | 9 (36) | 1.2 (0.6–7.5) | 14 (56) | 11 (44) | 1.7 (0.7–9.8) |
| Divorced | 7 (70) | 3 (30) | 1 | 6 (60) | 4 (40) | 1 |
| Residence | | | | | | |
| Rural | 185 (56) | 145 (44) | 1 | 204 (62) | 126 (38) | 1 |
| Urban | 188 (87) | 28 (13) | 4.3 (2.6–15.8)* | 164 (76) | 52 (24) | 1.7(1.1–12.4)* |
| Year of study | | | | | | |
| First | 129 (56) | 101 (44) | 1 | 145 (63) | 85 (34) | 1 |
| Second | 79 (61) | 50 (39) | 1.9 (0.7–10.3) | 74 (57) | 55 (43) | 2.6 (0.9–13.6) |
| Third | 80 (59) | 56(41) | 1.5 (0.8–8.2) | 90 (66) | 46 (34) | 1.7 (0.6–8.5) |
| Fourth and fifth years | 35 (68.6 | 16 (31.4) | 1.4 (1.1–12.5)* | 32 (62.7) | 19 (37.3) | 1.3 (0.5–8.5) |
| Monthly pocket money | | | | | | |
| <200 | 105 (63) | 62 (37) | 1 | 90(54) | 77(46) | 1 |
| 200–400 | 172 (76) | 54 (24) | 0.5 (0.2–5.6) | 138(61) | 88(39) | 2.3(0.8–11.2) |
| ≥401 | 104 (68) | 49 (32) | 1.6 (0.7–10.3) | 96(62.8) | 57(37.2) | 3.2(0.7–14.6) |
| Source of information | | | | | | |
| News media | 82 (71.3) | 33 (28.7) | 1.8 (1.2–9.2)* | 87 (75.6) | 28 (24.4) | 2.3 (1.6–8.7)* |
| Social media | 333 (83) | 69 (17) | 3.4 (1.9–13.7)* | 314 (78) | 88 (22) | 2.6 (2.1–11.8)* |
| Government websites | 7 (63.6) | 4 (36.4) | 0.6 (0.4–4.3) | 8 (72.7) | 3 (27.3) | 1.3 (1.1–6.2)* |
| Friends & family | 12 (66.7) | 6 (33.3) | 1 | 11 (61) | 7 (39) | 1 |

Remark:

* implies p< 0.05.

interventions and public health policies [12–15]. To the best of our knowledge, this is the first study in Ethiopia examining the KAP towards COVID-19 among university students. This survey was conducted during the very early stages of the epidemic in Ethiopia and showed that the majority of the study participants were knowledgeable about COVID-19. A similar finding were reported in China [13], Jordan [15] and India [12]. However, findings from Saudi Arabia (31.9%) [16], Middle Eastern Arabic countries (17.0%) [17], India (19.36%) [18], Tanzania [19] and Uganda [20] showed that most of the community had inadequate knowledge. This might be due to the late confirmed case report of COVID-19 in Ethiopia, which might provide time to know about the disease. Additionally, the devastating news reported about the disease and the WHO declaration of the disease as a pandemic due to its severe pathogenicity and communicability [7, 8] might also have increased the students' knowledge.

Additionally, as in a study in Tanzania [19] and Jordan [15], approximately 71.4% responded correctly, indicating that the main clinical symptoms of COVID-19 are fever, fatigue, dry cough, and shortness of breath. The high knowledge of recruited samples on the main clinical symptoms could be explained by various factors such as the seriousness of the disease and the effectiveness of different education programs in the region.

The vast majority (95%) of participants also said that there is currently no cure for COVID-19. We also found that more than 73.6% of the participants used social media as their main source of information about COVID-19. A similar result was also reported in Jordan [15]. This can help quickly disseminate important new information, relevant new scientific findings, share diagnostic, treatment, and follow-up protocols, as well as compare different approaches globally by removing geographic boundaries for the event [19].

In this study, the overall attitude of DBU students towards COVID-19 prevention was favorable in that they believed it would be controlled and could be prevented with good hygiene, which is in line with a study done at medical and non-medical university students [15]. Specifically, most of the participants (81.8%) believed that COVID-19 will eventually be controlled, and 83.8% had confidence that Ethiopia could win the battle against the virus. Similar to a finding in Syria [29], Pakistan [30] and Sudan [31], we also found that most of the study participants had poor practices regarding COVID-19 prevention. We found that approximately 48% did not cover their mouth and nose with an elbow or tissue while coughing or sneezing, and 56% did not maintain a social distance of at least one meter between themselves and anyone who was coughing or sneezing [27]. It might be due to the absence and /or poor quality of protective equipment and reservation from using protective equipment due to some discomforts. Thus, priority needs to be given to improve prevention practices parallel to awareness creation and making protective equipment available [20].

According to the present study, students who were 25 years old or older were 1.6 times more knowledgeable about COVID-19 than those less than 20 years old. Additionally, we found that those living in urban areas were almost four times more knowledgeable about the virus than those living in rural areas. This finding is in line with a study performed in Indonesia [32], the Nepalese [33], Sudan [31] and the South Wollo Zone [34]. This might be because those who came from urban areas have a high probability of access to different means of communication, including television and radio, which are the main channels used to teach the community about diseases [29].

Moreover, DBU students who heard most of their information from news media were 1.8 times more knowledgeable regarding COVID-19 compared to those who heard information from friends and family. There was a significant positive correlation between year of study and COVID-19 knowledge scores. A congruent result was reported in a study in Indonesia [32] and the South Wollo Zone [34]. This may be due to ease of access to readily updated information to most students via the internet and social media [34].

With respect to the attitude of DBU students regarding COVID-19, those who lived in an urban residence had a 1.7 times more favorable attitude towards COVID-19 prevention and control compared with those living in rural areas. Moreover, participants who heard information about the disease from government websites had a 1.3 times more favorable attitude towards COVID-19 preventive measures compared to those who heard information from friends and family. This might be because government media might provide reliable information to participants, and those who heard information from friends might hear rumors that may confuse them [12].

Despite our extensive efforts in reducing the possible shortcomings of this survey, this study does have certain limitations. First, the study was limited to the students of a government university only, and therefore, there is the question of the representativeness of the finding to all students. In addition, the data presented in this study are self-reported and thus may be subject to recall bias. Furthermore, the study also shares the limitations of establishing cause-effect relationships because of a cross-sectional study design.

## Conclusion and recommendation

This study summarized that DBU students had moderate knowledge about COVID-19 and positive attitudes toward tackling the disease. However, most participants had poor practices with respect to containing the spread of the disease. Our study also found that urban residence, year of study, age greater than 25, and source of information about COVID-19 were significantly associated with the mean knowledge and attitude of the study participants. Importantly, nearly half of the study participants did not cover their mouth and nose with an elbow or tissue while coughing and sneezing. Moreover, more than half of the study subjects did not maintain a physical distance of at least one meter between themselves and anyone who was coughing or sneezing. Therefore, an increased effort should be made to improve the knowledge and practices of DBU students regarding COVID-19 prevention. We also recommended that the Ministry of Sciences and Higher Education [MOSHE], a zonal health bureau in collaboration with universities, implement health education programs about corona virus to university students and the wider community to address a gap in KAP towards the preventive measures of COVID-19. Social media should also work by providing timely and reliable data to students about COVID-19 KAP. A large population-based study also needs to be conducted to assess the community KAP level at large. The university in collaboration with the zonal health bureau also needs to prepare a rule a regulation on covering mouth and nose with an elbow or tissue while coughing and sneezing and to maintain a physical distancing.

## Supporting information

**S1 File. A table of relevant demographic details.**
(DOCX)

**S2 File. Questioner used to assess the KAP of students.**
(DOCX)

## Acknowledgments

The authors thank all participants involved in this study for their cooperation and support.

## Author Contributions

**Conceptualization:** Yared Asmare Aynalem, Tadess Yirga Akalu, Birhan Gebresellassie Gebregiorgis, Nigussie Tadesse Sharew.

**Data curation:** Yared Asmare Aynalem, Nigussie Tadesse Sharew, Hilina Ketema Assefa.

**Formal analysis:** Yared Asmare Aynalem, Tadess Yirga Akalu, Birhan Gebresellassie Gebregiorgis, Nigussie Tadesse Sharew.

**Funding acquisition:** Yared Asmare Aynalem, Hilina Ketema Assefa.

**Investigation:** Tadess Yirga Akalu, Birhan Gebresellassie Gebregiorgis, Hilina Ketema Assefa, Wondimeneh Shibabaw Shiferaw.

**Methodology:** Tadess Yirga Akalu, Birhan Gebresellassie Gebregiorgis, Nigussie Tadesse Sharew, Hilina Ketema Assefa, Wondimeneh Shibabaw Shiferaw.

**Project administration:** Hilina Ketema Assefa.

**Resources:** Nigussie Tadesse Sharew, Hilina Ketema Assefa, Wondimeneh Shibabaw Shiferaw.

**Software:** Yared Asmare Aynalem, Hilina Ketema Assefa, Wondimeneh Shibabaw Shiferaw.

**Supervision:** Yared Asmare Aynalem, Wondimeneh Shibabaw Shiferaw.

**Validation:** Yared Asmare Aynalem, Wondimeneh Shibabaw Shiferaw.

**Visualization:** Yared Asmare Aynalem.

**Writing – original draft:** Yared Asmare Aynalem.

**Writing – review & editing:** Yared Asmare Aynalem, Wondimeneh Shibabaw Shiferaw.

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
