## [Decision Letter · Decision Letter 0]

10 Jul 2020

PONE-D-20-14158

Assessment of undergraduate student knowledge, practices, and attitude towards COVID-19 in Debre Berhan University, Ethiopia

PLOS ONE

Dear Dr. Aynalem,

Thank you for submitting your manuscript to PLOS ONE. After careful consideration, we feel that it has merit but does not fully meet PLOS ONE’s publication criteria as it currently stands. Therefore, we invite you to submit a revised version of the manuscript that addresses the points raised during the review process.

The reviewers have appreciated the urgency required to generate evidence regarding an emerging global pandemic such as covid-19, and praise in particular the benefit of such data from African settings. However, they have raised key methodological concerns and in particular, I would urge to consider the reviewers' comments, concerns and suggestions regarding the methodological aspects of the study, and the required improvement in the reporting of the conduct (sampling, selection, etc) of the research. 

I would also like you to appraise the comments offered by Reviewer 4 regarding some of the analytical decisions that you have made in the paper, and ensure that a response to all the points raised by the Reviewer is included. 

Finally, I would suggest that attention to the novelty aspects of the paper are not really required, as PLOS ONE focuses on the technical quality of the submitted work. 

We look forward to receiving your revised manuscript.

Kind regards,

Enrique Castro-Sánchez

Academic Editor

PLOS ONE

Journal Requirements:

2.  In the Methods, please clarify that participants provided oral consent. Please also state in the Methods:

- Why written consent could not be obtained

- Whether the Institutional Review Board (IRB) approved use of oral consent

- How oral consent was documented

For more information, please see our guidelines for human subjects research: " ext-link-type="uri" xlink:type="simple">https://journals.plos.org/plosone/s/submission-guidelines#loc-human-subjects-research"

3. Please include additional information regarding the survey or questionnaire used in the study and ensure that you have provided sufficient details that others could replicate the analyses. For instance, if you developed a questionnaire as part of this study and it is not under a copyright more restrictive than CC-BY, please include a copy, in both the original language and English, as Supporting Information."  

4. We noticed minor instances of text overlap with the following previous publication(s), which need to be addressed:

(a) https://www.ijbs.com/v16p1745.pdf

(b) https://news.mb.com.ph/2020/02/06/who-crafts-global-strategic-plan-vs-ncov/

(c) https://en.wikipedia.org/wiki/Pandemic

The text that needs to be addressed involves the Introduction section.

In your revision please ensure you cite all your sources (including your own works), and quote or rephrase any duplicated text outside the methods section. Further consideration is dependent on these concerns being addressed."

5. In your Methods section, please provide additional information about the participant recruitment method and the demographic details of your participants. Please ensure you have provided sufficient details to replicate the analyses such as:  a) a table of relevant demographic details, d) a statement as to whether your sample can be considered representative of a larger population and c) a description of how participants were recruited."

6. Please provide a sample size and power calculation in the Methods, or discuss the reasons for not performing one before study initiation

7. Please amend your list of authors on the manuscript to ensure that each author is linked to an affiliation. Authors’ affiliations should reflect the institution where the work was done (if authors moved subsequently, you can also list the new affiliation stating “current affiliation:….” as necessary).

8. Please include a copy of Tables 1-5 which you refer to in your text.

Reviewers' comments:

Reviewer's Responses to Questions

**Comments to the Author**

1. Is the manuscript technically sound, and do the data support the conclusions?

Reviewer #1: Yes

Reviewer #2: Yes

Reviewer #3: Yes

Reviewer #4: No

Reviewer #5: Partly

2. Has the statistical analysis been performed appropriately and rigorously? 

Reviewer #1: I Don't Know

Reviewer #2: Yes

Reviewer #3: Yes

Reviewer #4: No

Reviewer #5: Yes

3. Have the authors made all data underlying the findings in their manuscript fully available?

Reviewer #1: Yes

Reviewer #2: Yes

Reviewer #3: Yes

Reviewer #4: Yes

Reviewer #5: No

4. Is the manuscript presented in an intelligible fashion and written in standard English?

Reviewer #1: Yes

Reviewer #2: No

Reviewer #3: Yes

Reviewer #4: Yes

Reviewer #5: Yes

5. Review Comments to the Author

Reviewer #1: PONE-D-20-14158

Assessment of undergraduate student knowledge, practices, and attitude towards COVID-19 in Debre Berhan University, Ethiopia

Many thanks for the opportunity to review this paper.

There is a gap for the attitude to the pandemic, especially in Africa on the one hand, but also a flood of publications without a whole lot of purpose. I am slightly undecided whether this article enhances knowledge. Inclined to think to be publishable it needs a follow up element perhaps now or in 6 months time?

Abstract

I feel like the abstract doesn’t really give us key information on the findings – all I get from it is that some were less knowledgeable and there is a concern about educating them. Is that it? Hoping more specific as I read article.

Background

Some of the data is now 2 months old and outdated.. but I guess you had to draw line somewhere.

Repeated this line twice “seasonal flu generally kills far fewer than 1% of those infected[9]”

The Chinese study [10] was it before/ during/ after the surge in China as to whether comparable to your data (before).

I’m not sure quoting all the WHO strategies is helpful/ necessary. The whole 2nd last paragraph of the background is very much public knowledge at this stage and I don’t believe adds much.

Methods.

What is the intended faculty/ field of study of the students??

Questionnaire seems appropriate and rigorous.

Results

The field of study of the students?? I think this is vital. I would expect science/ health students to know more? Far more interested in this than their gender and age myself.

No Table 1, 2 in the version I have been given to view.

The results are succinct. Fine. I guess interesting in retrospect in years to come especially. But not remarkable – mostly as I would expect from educated students at this stage of the pandemic.

Discussion

I find this pretty superficial and repeating the results. Not a whole lot of interpretation. How does this compare even to the Chinese study you mentioned??

You don’t really express an opinion even..

Concl Recom

Again you don’t really pull out what is novel/ surprising.

What about recommendations for repeating this research later?

All you give is rather vague “educate” messages. How? Social media seems to work??

Reviewer #2: Covid19 related research is timely and should be supported. The effort of the authors is therefore praise worth. However,

• There are some grammatical errors that should be corrected.

• At the discussion section, the authors should compare their findings with what others have written before.

• The recommendations on educational campaigns should be made more explicit.

• The reference section should be checked for consistency, e.g. some emboldens

On the whole, the manuscript can be published after some major corrections. The suggested areas for correction can be seen in the attachment.

Reviewer #3: The paper presents an interesting and relevant research about the association between knowledge, attitude and practices (KAP) of Debre Berhan University students, towards COVID-19. The abstract presents the basic information regarding background, methods, results and conclusions, but the main objective of the paper is missing. To clarify the main aim of the article an objective needs to be included. Considering that the (KAP) towards COVID-19 is a central element, the title of the paper could be modified to adjust to that: “Assessment of undergraduate student knowledge, attitude and practices towards COVID-19 in Debre Berhan University, Ethiopia”.

The background section provides the main contents regarding COVID-19. The study took place in March and for this reason references about COVID-19 are also from March-April. However, as far as possible, it could be positive to have more recent data regarding confirmed cases, number of deaths and the number of cases reported in Debre Berhan and Ethiopia.

Just a brief comment regarding the information provided in the third paragraph regarding the countries that have been reporting COVID-19 cases. In this explanation the number of countries provided is 2214, but it is not possible, please revise the data.

At the end of this section a direct link is made between one study in China and the (KAP) towards the disease. It could be relevant to provide more analysis about how this association is made in other studies, particularly because this relation is essential in your study.

In the sub-section “study design, setting, and population” concrete and basic information regarding the population and the setting where the field work took place is included. However, little information about the study design is mentioned. Introducing here some explanation showing your design would be relevant for the article.

The information regarding the study variables is clear, with the outcome variable and the two independent covariates. But there is no explanation establishing the relation between both types of variables. Establishing a link between both through a research question would be important, because later, a binary logistic regression analysis to select factors has been done.

Data collection tool and procedure is correct in general terms, but the following aspects need to be improved:

- The explanation about the questionnaire. There are two parts and the first one, based in socio-demographic variables, is clear. But at the beginning of the second one regarding KAP, it would be necessary to explain that there are three different type of questions and a different way to work with the data and after that, provide an explanation of the three of them.

- The explanation about the knowledge about COVID-19 is correct, but it is important to justify why you have established three options per question when you are developing later a binary regression analysis. It is true that at the end you work with two - as you classify “No” and “I don’t know” as zero and “Yes” as one- but explaining the reasons why you decide to work in that way would be necessary.

- In the case of attitudes towards COVID-19 a five-point Likert scale has been used (strongly agree, agree, neutral, disagree, and strongly disagree). However, at the end, and following previous studies, three categories (agree, neutral and disagree) have been established. First of all, it might be useful to specify which are these previous studies? and second, you need to clarify the content of each category. Rationally “strongly agree” and “agree” must be in the category “agree” and “disagree” and “strongly disagree” in the category “disagree”. This needs to be explained.

- Information regarding participants’ practices needs more development. Which kind of questions are addressed to assess the participants’ practices? Are there dichotomic or is a Likert scale used? How many questions are included in this case?

- Finally, a short explanation about how the questionnaire is administered could be introduced.

Data processing and analysis section contains different types of statistics’ analysis. In the case of the descriptive statistics a relation between study participants and the “relevant variables” is suggested but insufficient information is provided concerning these relevant variables.

In the case of the multivariable analysis, a binary logistic regression analysis to identify the factors associated with KAP is used. Later, the factors were selected using backward stepwise method, but there is no information regarding this process, the main factors selected for the study and how they have been selected.

Results are presented by each type of statistics’ analysis, including the main information and being very clear in that way. The discussion section provides complementary explanations to the results. Finally, regarding conclusions and recommendation, including the main objective of the paper at the beginning would be positive and after that, the argumentation can follow. The recommendation about educational campaigns that are included just at the end is too short, more information could be provided on this.

A final comment regarding the references: the title or sometimes the authors appear in bold in all of them. Please check all references and make sure that all of them follow the journal’s requirements in this matter.

Reviewer #4: 1. The information provided about the data analysis is insufficient. In particular, coding of outcome variables in the regression models is unclear.

2. The authors do not provide reasons for using multivariate analysis with binary dependent variable.

3. The stepwise method is widely criticized in the literature. I do not find it appropriate either.

4. The strict use of 5% limit for discussing significance is not in line with current consensus in the literature of statistical analysis.

Reviewer #5: Thanks for the opportunity to revise this paper.

In this research, the authors present research on the knowledge, practices, and attitude towards COVID-19 amongst undergraduate students in a university of Ethiopia.

My major concern with the article relates to the explanation of the sample obtained. There is a void of information at that stage, while the remaining, methods, variables, analysis, are correctly reported.

In particular, no description at all is provided on how the sample was selected. What population does the sample represent? if any. As it is described, the sample seems to be a convenient sample (people ‘around’ were selected), not a representative sample of those 11000 undergraduate students. What were the response rates? The authors mentioned exclusions, but who were they and what were the reasons and how many of them were actually excluded. Did the authors conduct any power calculation? The authors claimed that the sample was of limited size, but it was above 500. What is large or small depends on the purpose of the sample, but this remains unclear without a power calculation. Why the authors reported to be of limited size remains unclear.

What is missing is a good literature review and discussion of the state of the art in KAP against COVID19 in countries across the world, amongst different groups (health professionals, politicians, general population, young educated people [this study]) across countries and maybe within Ethiopia. The literature used by the authors remains thin, with a lot of focus and quite lengthy description of general advice by WHO. We should know why KAP is relevant in relation to other crises and what are the KAP levels in Ethiopia and other countries. I acknowledge that there is possibly many more papers published now, relative to the time of writing this paper. However, literature for other infectious disease crisis exist, such as for Ebola in Western Africa, and could be helpful to give more depth to this work. Note that this has implications for the Discussion, too.

The discussion is clearly too short, was quickly put together, but more importantly, no links are made to any other literature. Some results such as a lower awareness in students from rural areas or an age effect should be discussed in light of other literature out there.

Below I provide a few minor comments,

Abstract, does not include essential methodological features, such as the sample.

[Page 1, Introduction] ‘By comparison, seasonal flu generally kills far fewer than 1% of those infected[9]’ This is repeated in the paragraph above. Please avoid repetition

[Page 1, Introduction] ‘A study in China showed that most respondents were knowledgeable about COVID-19 and the vast majority of the participants also held an optimistic attitude towards the COVID-19 pandemic. Specifically, 90.8% believed that COVID-19 would be successfully controlled, and 97.1% had confidence that China could win the battle against the virus [10].’ Add more studies.

References 1, 4, 9 and 15 are incomplete or have typos. Please revise.

Tables are not attached to the submission, so not evaluated.

6. PLOS authors have the option to publish the peer review history of their article (what does this mean?). If published, this will include your full peer review and any attached files.

Reviewer #1: **Yes: **PW Hodkinson

Reviewer #2: **Yes: **Dr John Bosco Azigwe

Reviewer #3: **Yes: **Ramon Flecha

Reviewer #4: No

Reviewer #5: **Yes: **Jose Manuel Rodriguez-Llanes

---

## [Author Response · Author response to Decision Letter 0]

22 Jul 2020

Dear Academic Editor!

PLOS ONE

Response to Academic Editor and Reviewers

I am pleased to resubmit for publication a revision version of “Assessment of undergraduate student knowledge, practices, and attitude towards COVID-19 in Debre Berhan University, Ethiopia” for a review as original research in PLOS ONE

.The comments of the editor and the reviewers were highly insightful and enabled us to greatly improve the quality of our manuscript. Therefore, based on the editor’s and the reviewers’ concerns we have made extensive edition in our manuscript. The comments of the editor and the reviewers were highly insightful and enabled us to greatly improve the quality of our manuscript. Therefore, based on the editor’s and the reviewers’ concerns we have made extensive edition in our manuscript. Especially we have extensively edited the manuscript by a professional language editor, at Excision Editing (a fluent, native Australian, English-language speaker thoroughly edited the manuscript for language usage, spelling, and grammar) before submitting the revised version. The formatting of the text and document (text sizes and grammatical errors) were also edited. His name is called Dr. Ryan Bell (CEO and Chief Editor Excision Editing)

 In the following pages, we have addressed yours’ concerns in a point by point format. 

 We look forward to hearing from you at your earliest convenience. 

Thank you for your consideration of this manuscript! 

Kind regards,

Yared Asmare Aynalem.

On behalf of authors

PLOS ONE Decision: Revision required [PONE-D-20-14158] - [EMID:3076548e22979785]

Inbox x

PLOS ONE

PONE-D-20-14158

Assessment of undergraduate student knowledge, practices, and attitude towards COVID-19 in Debre Berhan University, Ethiopia

PLOS ONE

Dear Dr. Aynalem,

Thank you for submitting your manuscript to PLOS ONE. After careful consideration, we feel that it has merit but does not fully meet PLOS ONE’s publication criteria as it currently stands. Therefore, we invite you to submit a revised version of the manuscript that addresses the points raised during the review process.

Response: Thank you very much for allowing us to revise our manuscript. We have tried to response each concerns as much as possible. We have submitted a revised version of the manuscript.

The reviewers have appreciated the urgency required to generate evidence regarding an emerging global pandemic such as covid-19, and praise in particular the benefit of such data from African settings. However, they have raised key methodological concerns and in particular, I would urge to consider the reviewers' comments, concerns and suggestions regarding the methodological aspects of the study, and the required improvement in the reporting of the conduct (sampling, selection, etc) of the research. I would also like you to appraise the comments offered by Reviewer 4 regarding some of the analytical decisions that you have made in the paper, and ensure that a response to all the points raised by the Reviewer is included. Finally, I would suggest that attention to the novelty aspects of the paper are not really required, as PLOS ONE focuses on the technical quality of the submitted work. 

Response: Thank you very much for recognizing the urgency required to generate evidence regarding an emerging global pandemic of covid-19, and giving praise to us .really excuse for missing these important methodological issues from the main manuscript. For that, we have included the detail methodological descriptions of the current study. Dear editor please see the color change in the manuscript. 

Response: Thank you. We have included ‘Response to Reviewers', 'Revised Manuscript with Track Changes ad Manuscript separately 

Response: thanks for allowing us to update the financial disclosure .we have added to the cover later .We didn’t have any figure .

 Response: thank you for recommending depositing the laboratory protocols in protocols.io. But it is not applicable

We look forward to receiving your revised manuscript.

Kind regards,

Enrique Castro-Sánchez

Academic Editor

PLOS ONE. 

Response: Thank you .we has set it 

Journal Requirements:

p1. Please ensure that your manuscript meets PLOS ONE's style requirements, inc;luding those for file naming. The PLOS ONE style templates can be found at https://journals.plos.org/plosone/s/file?id=wjVg/PLOSOne_formatting_sample_main_body.pdf and https://journals.plos.org/plosone/s/file?id=ba62/PLOSOne_formatting_sample_title_authors_affiliations.pdf

Response: thank you very much. We have sent the revised manuscript as based on the plose one guideline and as per editor’s comment 

2. In the Methods, please clarify that participants provided oral consent. Please also state in the Methods:

- Why written consent could not be obtained ,Whether the Institutional Review Board (IRB) approved use of oral consent, How oral consent was documented.For more information, please see our guidelines for human subjects research: https://journals.plos.org/plosone/s/submission-guidelines#loc-human-subjects-research"

Response: Dear our editor, we would like to say very excuse for making you confused .it was as an error. We have taken written informed consent. It was stated as Ethical clearance for this study was obtained from the Institute of Medicine and College of Health Sciences, DBU. It was approved by the University ethical approval commute with the approval number of Ref. No.DBU.R.D136/02/12. Participants also gave a written informed consent prior to data collection after the participates are informed that their participation is completely voluntary and anonymous, if they decide to take part, still free to withdraw at any time, all information that is collected will kept strictly confidential, personal details such as phone number and address will not be revealed to people outside the research teams. Confidentiality of the study participants' information was maintained throughout the study by making the participants' information anonymous. (see also tools that we used ad attached as the supplmetary files).

3. Please include additional information regarding the survey or questionnaire used in the study and ensure that you have provided sufficient details that others could replicate the analyses. For instance, if you developed a questionnaire as part of this study and it is not under a copyright more restrictive than CC-BY, please include a copy, in both the original language and English, as Supporting Information." 

Response: thanks for allowing us to add it .We have attached both the Amharic and English version of questionnaire used in the study (see also tools that we used ad attached as the supplmetary files).

4. We noticed minor instances of text overlap with the following previous publication(s), which need to be addressed:

(a) https://www.ijbs.com/v16p1745.pdf

(b) https://news.mb.com.ph/2020/02/06/who-crafts-global-strategic-plan-vs-ncov/

(c) https://en.wikipedia.org/wiki/Pandemic.The text that needs to be addressed involves the Introduction section.In your revision please ensure you cite all your sources, and quote or rephrase any duplicated text outside the methods section. Further consideration is dependent on these concerns being addressed."

Response: thanks for your recommendation for editing a text overlap o the above papers. The comments of the editor were highly insightful and enabled us to greatly improve the quality of our manuscript. Therefore, based on the editor’s and the reviewers’ concerns we have made extensive edition in our manuscript. Especially we have extensively edited the manuscript by a professional language editor, at Excision Editing (a fluent, native Australian, English-language speaker thoroughly edited the manuscript for language usage, spelling, and grammar) before submitting the revised version. The formatting of the text and document (text sizes and grammatical errors) were also edited. We have also tried to cite all sources. 

5. In your Methods section, please provide additional information about the participant recruitment method and the demographic details of your participants. Please ensure you have provided sufficient details to replicate the analyses such as: a) a table of relevant demographic details, d) a statement as to whether your sample can be considered representative of a larger population and c) a description of how participants were recruited."

Response: big thanks dear our editor. We have included a table of relevant demographic details as a supplementary files stated as follows. We also included about sample representatives ad a description of how participants were recruited. See the color change 

A table of relevant demographic details,

Sr o Collage Departments Total under graduated student in each collage Included collage data collection(randomly selected) 

1 School of Computing Science

 o Information technology

o Information system

o Computer science

o Software Engineering 1231

 � 

2 College of Engineering

 o Electrical and Computer Engineering

o Mechanical engineering

o Civil engineering

o Chemical engineering

o Construction technology and management

o Industrial Engineering

o Survey Engineering

o Food Processing Engineering

 4286 � 

3 College of Health Science

 o Nursing

o Midwifery

o Health officer

o Pediatrics Nursing 

o Neonatal Nursing 

o Surgical Nursing

o Medical Laboratory Science

 616 

4 College of Medicine

 o Anesthesia

o Medicine

o Pharmacy 263 

5 College of Natural and Computational Science

 o Biology

o Chemistry

o Physics

o Mathematics

o Sport science

o Statistics

o Biotechnology

o Geology 1231 

6 College of Business and Economics

 o Management

o Economics

o Accounting and Finance

o Tourism Management

o Logistics and Supply Chain Management 

o Marketing Management 1257 � 

7 College of Agriculture and Natural Resource Science

 o Plant science

o Animal science

o Natural resource management

o Water resource and irrigation management 

o Horticulture

o Agricultural Economics 741 � 

8 College of Social Science and Humanities

 o Geography and Environmental Studies

o History and heritage management

o Sociology

o Psychology

o English language and literature 

o Amharic

o Civics and ethical education

o Journalism and Communication 

 1210 

9 College of Law Law 221 

10 College of Education 

o Technical Drawing /Summer/

o Business Education /Summer/

o Special Need Education / Summer/

 1235 

6. Please provide a sample size and power calculation in the Methods, or discuss the reasons for not performing one before study initiation.

Response: Dear editor again we are sorry for the inconvenience. We have added Sample size determination and sampling procedure (procurement procedures) as follows: The sample size was calculated by using single population proportion formula assuming: 95% confidence level, 3% margin of error, and, 50% proportion for KAP level. Since there is no any study in our countries we have used 50%. After a 10% non-response rate and 1.5 design effects, the final sample size was 634. Multi-stage sampling procedure was used to select the study participants. Sample size was allocated proportionally to get the required sample size from each selected Colleges. The sampling fraction is approximately equal to seven for all colleges. Accordingly, every 7th students were selected using a systematic random sampling technique from each college registrar office log-book. After that a dorm to dorm visit were used to get the sampled students. Since we have used50% proportion, our sample can be considered as a representative of a larger population. Students in each selected collage that were present during the data collection period and were able to give responses were included in the study. Those who had mental or physical disabilities and could not fill out the questionnaires (unwilling to participate in to the study) were excluded from the study. 

7. Please amend your list of authors on the manuscript to ensure that each author is linked to an affiliation. Authors’ affiliations should reflect the institution where the work was done (if authors moved subsequently, you can also list the new affiliation stating “current affiliation:….” as necessary).

Response: thanks for your recommendation for amending the list of authors on the manuscript. We have amended it ad we have ensured that that each author is linked to an affiliation.

Yared Asmare Aynalem1* Tadess Yirga Akalu2¶,, Birhan Gebresellassie1, Nigussie Tadesse Sharew1 ,Wondimeneh Shibabaw Shiferaw1¶

1Department of Nursing, College of Health Science, Debre Berhan University, Debre Berhan, Ethiopia

2Department of Nursing, College of Health Science, Debre Markos University, Debre Markos, Ethiopia

*Corresponding author: yaredasmare123@gmail.com

¶ these authors contributed equally to this work. 

 these authors also contributed equally to this work.

8. Please include a copy of Tables 1-5 which you refer to in your text.

Response: thanks for reminding to add the missed Tables 1-5.We have included them. See the color change 

Reviewers' comments:

Reviewer's Responses to Questions

Comments to the Author

1. Is the manuscript technically sound, and do the data support the conclusions?

Reviewer #1: Yes

Reviewer #2: Yes

Reviewer #3: Yes

Reviewer #4: No

Reviewer #5: Partly

Response: thank you very much.I think we have tried to modified it see the track change 

2. Has the statistical analysis been performed appropriately and rigorously?

Reviewer #1: I Don't Know

Reviewer #2: Yes

Reviewer #3: Yes

Reviewer #4: No

Reviewer #5: Yes

Response: thank you very much. We have try to revised it 

3. Have the authors made all data underlying the findings in their manuscript fully available?

Reviewer #1: Yes

Reviewer #2: Yes

Reviewer #3: Yes

Reviewer #4: Yes

Reviewer #5: No

Response: thank you very much. We have included the remaining tool (questioner ) as a supplementary files 

4. Is the manuscript presented in an intelligible fashion and written in standard English?

Reviewer #1: Yes

Reviewer #2: No

Reviewer #3: Yes

Reviewer #4: Yes

Reviewer #5: Yes

Response: thank you very much. The comments of the reviewer were highly insightful and enabled us to greatly improve the quality of our manuscript. Therefore, based on the editor’s and the reviewers’ concerns we have made extensive edition in our manuscript. Especially we have extensively edited the manuscript by a professional language editor, at Excision Editing (a fluent, native Australian, English-language speaker thoroughly edited the manuscript for language usage, spelling, and grammar) before submitting the revised version. The formatting of the text and document (text sizes and grammatical errors) were also edited. We have also tried to cite all sources. 

5. Review Comments to the Author

Response: thank you very much. We have tried to answer the reviewer’s feedback ad question as much as we can.

Reviewer #1: PONE-D-20-14158

Assessment of undergraduate student knowledge, practices, and attitude towards COVID-19 in Debre Berhan University, Ethiopia.Many thanks for the opportunity to review this paper.

There is a gap for the attitude to the pandemic, especially in Africa on the one hand, but also a flood of publications without a whole lot of purpose. I am slightly undecided whether this article enhances knowledge. Inclined to think to be publishable it needs a follow up element perhaps now or in 6 months time?

Response: thank you very much for reviewing the paper. As far as we know, still there is scarcity of data particularly in low income countries including Ethiopia. This data may also help for the future systematic review ad meta-analysis to synthesis results.

Abstract

I feel like the abstract doesn’t really give us key information on the findings – all I get from it is that some were less knowledgeable and there is a concern about educating them. Is that it? Hoping more specific as I read article.

Response: We thank you very much for allowing us to improve the abstract of this finding. We have tried to improve it. See the color change please. 

Background

Some of the data is now 2 months old and outdated.. but I guess you had to draw line somewhere. Repeated this line twice “seasonal flu generally kills far fewer than 1% of those infected[9]” The Chinese study [10] was it before/ during/ after the surge in China as to whether comparable to your data (before). I’m not sure quoting all the WHO strategies is helpful/ necessary. The whole 2nd last paragraph of the background is very much public knowledge at this stage and I don’t believe adds much.

Response: We thank you very much for the concern. We have used the latest /updated report of data as per your constructive feedback .thank you for that. We have also removed the repeated sentence. We have also modified the WHO strategies which is helpful to compate COVID-19 currently 

Methods.

What is the intended faculty/ field of study of the students?? Questionnaire seems appropriate and rigorous.

Response: Thank you. The source population of this study was all undergraduate students at Debre Berhan University and the study population was all students in the randomly selected collages (Collage of engendering, computing science, Faculty of business and economics and collage of agriculture).

Results

The field of study of the students?? I think this is vital. I would expect science/ health students to know more? Far more interested in this than their gender and age myself.

Response: since we have selected 4 collages randomly (Collage of engendering, computing science, Faculty of business and economics and collage of agriculture) we didn’t report each separately. because the finding of this study is generalized to all collages.

No Table 1, 2 in the version I have been given to view.

The results are succinct. Fine. I guess interesting in retrospect in years to come especially. But not remarkable – mostly as I would expect from educated students at this stage of the pandemic.

Response: thanks for reminding to add the missed Tables 1-5.We have included them. See the color change 

Discussion

I find this pretty superficial and repeating the results. Not a whole lot of interpretation. How does this compare even to the Chinese study you mentioned??You don’t really express an opinion even..

Response: thanks .We have rewrite the overall parts of the discussion by comparing with differ finding to the world. Please see the color change 

Concl Recom

Again you don’t really pull out what is novel/ surprising. What about recommendations for repeating this research later?All you give is rather vague “educate” messages. How? Social media seems to work??

Response: thanks for asking us to improve the conclusion and recommendation. We have revised this part intensively. We have tried to make our recommendations as specific as possible See the color change.

Reviewer #2: Covid19 related research is timely and should be supported. The effort of the authors is therefore praise worth. 

Response: Thank you for acknowledging that 

However,

• There are some grammatical errors that should be corrected.

Response: Thank you. The comments of the reviewer were highly insightful and enabled us to greatly improve the quality of our manuscript. Therefore, based on the editor’s and the reviewers’ concerns we have made extensive edition in our manuscript. Especially we have extensively edited the manuscript by a professional language editor, at Excision Editing (a fluent, native Australian, English-language speaker thoroughly edited the manuscript for language usage, spelling, and grammar) before submitting the revised version. The formatting of the text and document (text sizes and grammatical errors) were also edited. 

• At the discussion section, the authors should compare their findings with what others have written before.

Response: Thank you for your golden feedback. We have discussed it as per your suggestions 

• The recommendations on educational campaigns should be made more explicit.

Response: Thank you. We have tried to make our recommendations as specific as possible See the color change.

• The reference section should be checked for consistency, e.g. some emboldens

Response: Thank you. We have revised all part of the reference 

Reviewer #3: The paper presents an interesting and relevant research about the association between knowledge, attitude and practices (KAP) of Debre Berhan University students, towards COVID-19. The abstract presents the basic information regarding background, methods, results and conclusions, but the main objective of the paper is missing. To clarify the main aim of the article an objective needs to be included. Considering that the (KAP) towards COVID-19 is a central element, the title of the paper could be modified to adjust to that: “Assessment of undergraduate student knowledge, attitude and practices towards COVID-19 in Debre Berhan University, Ethiopia”.

Response: Thank you for acknowledging that. We have included an objective at the abstract part 

The background section provides the main contents regarding COVID-19. The study took place in March and for this reason references about COVID-19 are also from March-April. However, as far as possible, it could be positive to have more recent data regarding confirmed cases, number of deaths and the number of cases reported in Debre Berhan and Ethiopia.

Response: Thank you. We have included the most recent data as per your suggestion. See the color change. 

Just a brief comment regarding the information provided in the third paragraph regarding the countries that have been reporting COVID-19 cases. In this explanation the number of countries provided is 2214, but it is not possible, please revise the data.

Response: Thank you. We have included the most recent data as per your suggestion(icludig worldometer report). See the color change. 

At the end of this section a direct link is made between one study in China and the (KAP) towards the disease. It could be relevant to provide more analysis about how this association is made in other studies, particularly because this relation is essential in your study.

Response: Thank you for your golden feedback. We have discussed it as per your suggestions. See the color change.

In the sub-section “study design, setting, and population” concrete and basic information regarding the population and the setting where the field work took place is included. However, little information about the study design is mentioned. Introducing here some explanation showing your design would be relevant for the article.

Response: Thank you for you. A cross-sectional study was conducted from March 18–24, 2020, with undergraduate DBU students. For more detail please also see the color change. 

The information regarding the study variables is clear, with the outcome variable and the two independent covariates. But there is no explanation establishing the relation between both types of variables. Establishing a link between both through a research question would be important, because later, a binary logistic regression analysis to select factors has been done.

Data collection tool and procedure is correct in general terms, but the following aspects need to be improved:

Response: Thank you :In this study, knowledge (good/poor), attitude (positive/negative), practice (good/poor) towards COVID-19 was the outcome variable. Socio-demographics factor like age, residence, sex, marital status, educational level, field of study, income, family size, religion) and the source of information regarding COVID-19 were considered the independent covariates. Knowledge for attitude, knowledge and attitude for practice towards the preventive measures of COVID-19 were also assessed. Correlation between independent variables (including the correlation between knowledge, attitude ad practice) was assessed .But we did not find any Correlation

- The explanation about the questionnaire. There are two parts and the first one, based in socio-demographic variables, is clear. But at the beginning of the second one regarding KAP, it would be necessary to explain that there are three different type of questions and a different way to work with the data and after that, provide an explanation of the three of them.

Response: Thank you for your feedback .We have modified it as follows: The questionnaire consisted of four parts: 1) socio-demographics; 2) knowledge; 3) attitudes; and 4) practices of to COVID-19. Socio-demographic variables included age, sex, and marital status, and residence, year of study, average monthly income, and source of information.

- The explanation about the knowledge about COVID-19 is correct, but it is important to justify why you have established three options per question when you are developing later a binary regression analysis. It is true that at the end you work with two - as you classify “No” and “I don’t know” as zero and “Yes” as one- but explaining the reasons why you decide to work in that way would be necessary.

Response: thank you dear reviewer r .due to the small number of participates responding as in don’t know, we have recatagorized.

- In the case of attitudes towards COVID-19 a five-point Likert scale has been used (strongly agree, agree, neutral, disagree, and strongly disagree). However, at the end, and following previous studies, three categories (agree, neutral and disagree) have been established. First of all, it might be useful to specify which are these previous studies? and second, you need to clarify the content of each category. Rationally “strongly agree” and “agree” must be in the category “agree” and “disagree” and “strongly disagree” in the category “disagree”.

Response: thank you dear reviewer. We have cited the previous study that we used as a base line. strongly agree and agree were category as “agree” and “disagree” and “strongly disagree” in the category of “disagree. The additional rational is due to the small number of participate.

- Information regarding participants’ practices needs more development. Which kind of questions are addressed to assess the participants’ practices? Are there dichotomic or is a Likert scale used? How many questions are included in this case?

- Finally, a short explanation about how the questionnaire is administered could be introduced.

Response: thank you dear reviewer. We have used dichotomic to assess the participants’ practices based on the who tools and different finding that we have cited .See also the color change .

Data processing and analysis section contains different types of statistics’ analysis. In the case of the descriptive statistics a relation between study participants and the “relevant variables” is suggested but insufficient information is provided concerning these relevant variables.

Response: thank you dear reviewer. Sorry for making you confused. We have made a correction (to show variables of Socio-demographic characteristics, participants’ knowledge, and attitude ad practice regarding COVID-19).

In the case of the multivariable analysis, a binary logistic regression analysis to identify the factors associated with KAP is used. Later, the factors were selected using backward stepwise method, but there is no information regarding this process, the main factors selected for the study and how they have been selected.

Results are presented by each type of statistics’ analysis, including the main information and being very clear in that way. 

The discussion section provides complementary explanations to the results. Finally, regarding conclusions and recommendation, including the main objective of the paper at the beginning would be positive and after that, the argumentation can follow. The recommendation about educational campaigns that are included just at the end is too short, more information could be provided on this.

A final comment regarding the references: the title or sometimes the authors appear in bold in all of them. Please check all references and make sure that all of them follow the journal’s requirements in this matter.

Reviewer #4: 1. The information provided about the data analysis is insufficient. In particular, coding of outcome variables in the regression models is unclear.

Response: we the authors would like to thanks for your valuable feedback. We tried to modified it as sufficient as possible per your suggestion (see the color change).the outcome variables were coded as follows: Coding of outcome variables in the regression models were stated as follows. 1 point was given to adequate, and 0 inadequate in the Knowledge; 1 for positive, and 0 for negative in the Attitude; 1 for good, and 0 for poor in the practice part.

2. The authors do not provide reasons for using multivariate analysis with binary dependent variable.

Response: Very excuse dear reviewer it is a typing error. It is to mean Multivariable regression results of the binary logistic regression analyses that control cofounding, not to mean multivariate analysis.

3. The stepwise method is widely criticized in the literature. I do not find it appropriate either.

Response :dear reviewer we acknowledge your feedback 

Certain regression selection approaches are helpful in testing predictors, thereby increasing the efficiency of analysis. Therefore, we thanks for allowing us to select the best regression selection approaches.we revised ad reanalysis using the default (Entry Method), which is the standard method of entry is simultaneous ( the enter method); all independent variables are entered into the equation at the same time. Each predictor is assessed as though it were entered after all the other independent variables were entered, and assessed by what it offers to the prediction of the dependent variable that is different from the predictions offered by the other variables entered into the model.

4. The strict use of 5% limit for discussing significance is not in line with current consensus in the literature of statistical analysis.

Response: dear reviewer thank you for your up to dated evidence of it. but we didn’t got any new evidence that supports to declare significance .We have tried to read more including evidences entitled Moving to a World Beyond “p 0.05” y Ronald L. Wasserstein, Allen L. Schirm Nicole A. Lazar. Still they suggested importantly, when reporting p values, authors should always provide the actual value, not only statements of “p 0.05” or “p ≥ 0.05”, because p values give a measure of the degree of data compatibility with the null hypothesis. Please suggest if any 

Reviewer #5: Thanks for the opportunity to revise this paper.

In this research, the authors present research on the knowledge, practices, and attitude towards COVID-19 amongst undergraduate students in a university of Ethiopia.

My major concern with the article relates to the explanation of the sample obtained. There is a void of information at that stage, while the remaining, methods, variables, analysis, are correctly reported. In particular, no description at all is provided on how the sample was selected. What population does the sample represent? if any. As it is described, the sample seems to be a convenient sample (people ‘around’ were selected), not a representative sample of those 11000 undergraduate students. What were the response rates? The authors mentioned exclusions, but who were they and what were the reasons and how many of them were actually excluded. Did the authors conduct any power calculation? The authors claimed that the sample was of limited size, but it was above 500. What is large or small depends on the purpose of the sample, but this remains unclear without a power calculation. Why the authors reported to be of limited size remains unclear.

Response :thank you .the sampling methods are modified as follows:

The sample size was calculated by using single population proportion formula assuming: 95% confidence level, 3% margin of error, and, 50% proportion for KAP level. Since there is no any study in our countries we have used 50%. After a 10% non-response rate and 1.5 design effects, the final sample size was 634. Multi-stage sampling procedure was used to select the study participants. Sample size was allocated proportionally to get the required sample size from each selected Colleges. The sampling fraction is approximately equal to seven for all colleges. Accordingly, every 7th students were selected using a systematic random sampling technique from each college registrar office log-book. After that a dorm to dorm visit were used to get the sampled students. Since we have used50% proportion, our sample can be considered as a representative of a larger population. The response rate of 86.1%.

.we has corrected all the mistakes. See the color change 

What is missing is a good literature review and discussion of the state of the art in KAP against COVID19 in countries across the world, amongst different groups (health professionals, politicians, general population, young educated people [this study]) across countries and maybe within Ethiopia. The literature used by the authors remains thin, with a lot of focus and quite lengthy description of general advice by WHO. We should know why KAP is relevant in relation to other crises and what are the KAP levels in Ethiopia and other countries. I acknowledge that there is possibly many more papers published now, relative to the time of writing this paper. However, literature for other infectious disease crisis exist, such as for Ebola in Western Africa, and could be helpful to give more depth to this work. Note that this has implications for the Discussion, too.

Response: thank you for your feedback. We have extensively edited the entire paper including the introduction by including the latest data. Please see the color change 

The discussion is clearly too short, was quickly put together, but more importantly, no links are made to any other literature. Some results such as a lower awareness in students from rural areas or an age effect should be discussed in light of other literature out there.

Response: thank you. We have addressed this issue dear reviewer . See the color change 

Below I provide a few minor comments,

Abstract, does not include essential methodological features, such as the sample.

Response: thank you. We have addressed this issue. See the change 

[Page 1, Introduction] ‘By comparison, seasonal flu generally kills far fewer than 1% of those infected[9]’ This is repeated in the paragraph above. Please avoid repetition

[Page 1, Introduction] ‘A study in China showed that most respondents were knowledgeable about COVID-19 and the vast majority of the participants also held an optimistic attitude towards the COVID-19 pandemic. Specifically, 90.8% believed that COVID-19 would be successfully controlled, and 97.1% had confidence that China could win the battle against the virus [10].’ Add more studies.

References 1, 4, 9 and 15 are incomplete or have typos. Please revise.

Tables are not attached to the submission, so not evaluated.

Response: thank you big thanks. We revised the reference ad we think that we have modified. Additionally, we have deleted any redundancy and we have tried to use the latest data .thank you for your golden feedback

6. PLOS authors have the option to publish the peer review history of their article (what does this mean?). If published, this will include your full peer review and any attached files.

Do you want your identity to be public for this peer review? For information about this choice, including consent withdrawal, please see our Privacy Policy.

Reviewer #1: Yes: PW Hodkinson

Reviewer #2: Yes: Dr John Bosco Azigwe

Reviewer #3: Yes: Ramon Flecha

Reviewer #4: No

Reviewer #5: Yes: Jose Manuel Rodriguez-Llanes

Dear all, it’s our gratitude for all of you for spending your valuable time to our paper. It helps us to make strong. Thank you

Stay safe

---

## [Decision Letter · Decision Letter 1]

27 Aug 2020

PONE-D-20-14158R1

Assessment of undergraduate student knowledge, practices, and attitude towards COVID-19 in Debre Berhan University, Ethiopia

PLOS ONE

Dear Dr. Aynalem,

Thank you for submitting your manuscript to PLOS ONE. After careful consideration, we feel that it has merit but does not fully meet PLOS ONE’s publication criteria as it currently stands. Therefore, we invite you to submit a revised version of the manuscript that addresses the points raised during the review process.

The reviewers have very much appreciated the effort the authors have made in considering and introducing the suggestion proposed- however, they still find the language unclear and in need of another grammatical revision. I understand that a professional proofreader reviewed the paper, but several sentences across the paper remain below an acceptable level of clarity. One of the reviewers identified so many unclear sentences and phrases that felt frustrated, and this may be something the readers may feel as well.

Additionally, please note the comments by the reviewers about the new information introduced regarding the sampling strategy, and I would urge you to examine the responses offered previously about the justification for such sampling strategy, as I felt such response was not as detailed and robust as ideal.

Finally, consider the suggestion by the reviewers to evaluate the Discussion and recommendations, as they remain short in length and unconnected to the aims and results of the paper, as well as existing literature.

As PLOS ONE focuses on scientific quality rather than novelty or impact, consideration towards the clarity, thoroughness and addition to the existing body of evidence in the topic of the paper become even more fundamental.

We look forward to receiving your revised manuscript.

Kind regards,

Enrique Castro-Sánchez

Academic Editor

PLOS ONE

Reviewers' comments:

Reviewer's Responses to Questions

**Comments to the Author**

1. If the authors have adequately addressed your comments raised in a previous round of review and you feel that this manuscript is now acceptable for publication, you may indicate that here to bypass the “Comments to the Author” section, enter your conflict of interest statement in the “Confidential to Editor” section, and submit your "Accept" recommendation.

Reviewer #1: (No Response)

Reviewer #2: All comments have been addressed

Reviewer #3: All comments have been addressed

Reviewer #4: All comments have been addressed

Reviewer #5: (No Response)

2. Is the manuscript technically sound, and do the data support the conclusions?

Reviewer #1: Yes

Reviewer #2: Yes

Reviewer #3: Yes

Reviewer #4: Yes

Reviewer #5: Partly

3. Has the statistical analysis been performed appropriately and rigorously? 

Reviewer #1: Yes

Reviewer #2: Yes

Reviewer #3: Yes

Reviewer #4: Yes

Reviewer #5: No

4. Have the authors made all data underlying the findings in their manuscript fully available?

Reviewer #1: Yes

Reviewer #2: Yes

Reviewer #3: Yes

Reviewer #4: Yes

Reviewer #5: No

5. Is the manuscript presented in an intelligible fashion and written in standard English?

Reviewer #1: Yes

Reviewer #2: Yes

Reviewer #3: Yes

Reviewer #4: Yes

Reviewer #5: No

6. Review Comments to the Author

Reviewer #1: Thankyou much improved

I can see you have put a lot of work in and i think it is useful now.

Still riddled with grammar and language issues.

I started pointing out and then got frustrated so below list is not complete. Needs proofreading and correctig again please.

Methods

sample size first paragraph:

Since there is no any study in our countries - correct

10%v non response rate?? That’s unheard of for most surveys….

Spelling collage - college

Since there is no any study in our countries

Results

Table 1 can you indicate the correct answers somehow?

Reviewer #2: The grammatical errors have been largely addressed. In the recommendation section, the authors should improve to include other dissemination strategies that can be more effective in rural context such as the study sight. See attachment

Reviewer #3: Dear authors,

You have made a great effort with this new article submission. You have included in your submission a marked-up copy of your manuscript highlighting changes and a rebuttal letter responding also to each point raised by the editor and reviewers. Both documents show clearly your new work based in all editor and reviewers’ previous observations. You have improved especially theoretical background and methodological aspects and analysis following our suggestions.

The information regarding written informed consent for your participants and the explanation regarding ethical issues is now clear. The questionnaire in English is so helpful to understand more in depth the results. Following the suggestion of the editor and reviewers you have made extensive edition by a native speaker.

The abstract has been improved following the reviewers’ suggestions. Authors have included the main objective of the article and now there is information regarding findings also in the abstract.

The theoretical background has now new references giving the paper more consistency and comparativeness with other regional areas. Now you have used the latest report of data including different regions and you have modified the WHO strategies.

In the Methods section there is new information answering many diverse comments from the reviewers. You have improved this section correctly, especially regarding participant recruitment method and demographic details. Tables about participants and the recruitment help in this way. The explanation on sample size and power calculation is also clear now. Information regarding how the variables were coded is now improved and the clarification about the multivariable regression instead of multivariate analysis is now clear. The information about the questionnaire is now improved and uploaded in the system. The discussion has now more in-depth analysis, including different finding around the world linked with your results.

The conclusion and recommendation section needs more development. You have improved the conclusions, but it is to short and the recommendations are still so vague. You could start this section confirming (or not) if your main objective (included now in the abstract) has been reached with your research work. Later you could also comment if you have found some correlations or not in your work. Finally, you could provide more details in your recommendations.

Reviewer #4: The authors have addressed my critiques regarding the presentation and analysis of the data. The revised presentation of the smaple, variables and tha analysis is clear and apropriate.

Reviewer #5: The KAP Survey Model stands for Knowledge, Attitudes, and Practices. The title may reflect this order. Also it is more logical.

The manuscript lacks a more convincing argumentation on why COVID19' KAP is important amongst students. In other words, why to study that population and not another one, or the general population, where poverty is expected to be more prevalent and interventions high priority.

In Methods (section entitled ‘Sample size determination and sampling procedure (procurement procedures’)

The authors wrote as new text ‘Since we have used 50% proportion, our sample can be considered as a representative of a larger population’. This sentence makes no sense and the authors should remove it from the manuscript. Setting a 50% level for each of the variables in the survey is helpful as it is a conservative approach when no previous knowledge is available on expected levels of a variable within a population. Setting that level ensures that the sample size will be sufficient for any level as a 50% prevalence or proportion generates the highest requirement for sample, keeping other known parameters constant.

Importantly, the authors mentioned several times (in manuscript and reply to reviewers) that they intended to represent the student population in that university. If that is so, then the described two stage sampling strategy selected by the authors is not self-weighted, and the authors need to calculate weights and incorporate those into their analysis. In addition there is attrition from the 'planned' sample to the 'collected' sample but little detail on circumstances (not present in dormitory at time of interview, refused to participate, provided incomplete answers or partial answers).

In the Discussion section, the authors wrote the following new text:

‘A supportive results were reported in China [13],Jordan [15] and India [12]. However, finding from Saudi Arabia 31.9%[16], Middle Eastern Arabic countries 17.0%[17],India 19.36% [18],in Tanzania [19] and in Uganda [20] showed that most of the community had inadequate Knowledge .This might be due to the late confirmed case report of COVID-19 in Ethiopia which might gave a time to know about the disease. Additionally, the devastating news reported about the disease, and the WHO declaration of the disease as a pandemic due to its sever pathogenicity and communicability [7, 8]. might also have increased the students’ knowledge.’

The first sentence above is not good English but the authors claimed the paper to be edited and polished by professionals. It does not seem to be the case here. Another example is below.

‘This finding is in line with a study done in Indonesian[31], Nepalese[32], Sudan[30] and South Wollo Zone[33]’.

Overall, the discussion is gaining some importance and becoming more balanced but requires better expression of the main ideas, and improving clarity. I see many punctuation errors.

References no. 1, 10 and 11, seem incomplete.

7. PLOS authors have the option to publish the peer review history of their article (what does this mean?). If published, this will include your full peer review and any attached files.

Reviewer #1: **Yes: **PW Hodkinson

Reviewer #2: **Yes: **Dr John Bosco Azigwe

Reviewer #3: **Yes: **Ramon Flecha

Reviewer #4: No

Reviewer #5: **Yes: **Jose Manuel Rodriguez-Llanes

---

## [Author Response · Author response to Decision Letter 1]

28 Aug 2020

Dear Academic Editor!

Plose one 

Response to Academic Editor 

I am pleased to resubmit(R2) for publication version of “Assessment of undergraduate student knowledge, practices, and attitude towards COVID-19 in Debre Berhan University, Ethiopia for a review as original research in Plose one 

.The comments of the editor and reviewer were highly insightful and enabled us to greatly improve the quality of our manuscript. Therefore, based on the editor and reviewer concerns we have made extensive edition in our manuscript. The comments of the editor and reviewer r were highly insightful and enabled us to greatly improve the quality of our manuscript. Therefore, based on the editor and reviewer concerns we have made extensive edition in our manuscript. We have edited using the free editing chance that is given to researcher until august 31 at Research Square is currently offering free use of an artificial intelligence editing tool during the Beta period called Automated Editing tool using the following links https://www.researchsquare.com/static/pdf/automated-editing-tool-instructions.pdf . Additionally, we have extensively edited the manuscript by a professional language editor, at Excision Editing (a fluent, native Australian, English-language speaker thoroughly edited the manuscript for language usage, spelling, and grammar) before submitting the revised version. The formatting of the text and document (text sizes and grammatical errors) were also edited. His name is called Dr. Ryan Bell(CEO and Chief Editor Excision Editing)

 In the following pages, we have addressed yours’ concerns in a point by point format. 

 We look forward to hearing from you at your earliest convenience. 

Thank you for your consideration of this manuscript! 

Kind regards,

Yared Asmare Aynalem.

On behalf of authors

Editor’s comment 

ONE-D-20-14158R1

Assessment of undergraduate student knowledge, practices, and attitude towards COVID-19 in Debre Berhan University, Ethiopia

PLOS ONE

Dear Dr. Aynalem,

Thank you for submitting your manuscript to PLOS ONE. After careful consideration, we feel that it has merit but does not fully meet PLOS ONE’s publication criteria as it currently stands. Therefore, we invite you to submit a revised version of the manuscript that addresses the points raised during the review process.

Response: Thank you very much for allowing us to revise our manuscript again. We have tried to response each concerns as much as possible. We have submitted a revised version of the manuscript.

The reviewers have very much appreciated the effort the authors have made in considering and introducing the suggestion proposed- however, they still find the language unclear and in need of another grammatical revision. I understand that a professional proofreader reviewed the paper, but several sentences across the paper remain below an acceptable level of clarity. One of the reviewers identified so many unclear sentences and phrases that felt frustrated, and this may be something the readers may feel as well.

Response: Thank you very much. We have edited it again using the free editing chance that is given to researcher until august 31 at Research Square is currently offering free use of an artificial intelligence editing tool during the Beta period called Automated Editing tool using the following links https://www.researchsquare.com/static/pdf/automated-editing-tool-instructions.pdf . 

Additionally, please note the comments by the reviewers about the new information introduced regarding the sampling strategy, and I would urge you to examine the responses offered previously about the justification for such sampling strategy, as I felt such response was not as detailed and robust as ideal. Finally, consider the suggestion by the reviewers to evaluate the Discussion and recommendations, as they remain short in length and unconnected to the aims and results of the paper, as well as existing literature.

Response: thanks our respected editor .We the authors tried to address all the concerns as much as we ca. please see the track change 

As PLOS ONE focuses on scientific quality rather than novelty or impact, consideration towards the clarity, thoroughness and addition to the existing body of evidence in the topic of the paper become even more fundamental. Please submit your revised manuscript by Oct 11 2020 11:59PM. If you will need more time than this to complete your revisions, please reply to this message or contact the journal office at plosone@plos.org. Please include the following items when submitting your revised manuscript:

• Response: Thank you. We have included ‘Response to Reviewers', 'Revised Manuscript with Track Changes ad Manuscript separately 

We look forward to receiving your revised manuscript.

Kind regards,

Enrique Castro-Sánchez

Academic Editor

PLOS ONE

Reviewers' comments:

Reviewer's Responses to Questions

Comments to the Author

1. If the authors have adequately addressed your comments raised in a previous round of review and you feel that this manuscript is now acceptable for publication, you may indicate that here to bypass the “Comments to the Author” section, enter your conflict of interest statement in the “Confidential to Editor” section, and submit your "Accept" recommendation.

Reviewer #1: (No Response)

Reviewer #2: All comments have been addressed

Reviewer #3: All comments have been addressed

Reviewer #4: All comments have been addressed

Reviewer #5: (No Response)

Response: thank you for acknowledging that 

2. Is the manuscript technically sound, and do the data support the conclusions?

Reviewer #1: Yes

Reviewer #2: Yes

Reviewer #3: Yes

Reviewer #4: Yes

Reviewer #5: Partly

Response: thank you for acknowledging that .we the authors tried to address it again 

3. Has the statistical analysis been performed appropriately and rigorously?

Reviewer #1: Yes

Reviewer #2: Yes

Reviewer #3: Yes

Reviewer #4: Yes

Reviewer #5: No

Response: thank you for acknowledging that .we the authors tried to address it again 

4. Have the authors made all data underlying the findings in their manuscript fully available?

Reviewer #1: Yes

Reviewer #2: Yes

Reviewer #3: Yes

Reviewer #4: Yes

Reviewer #5: No

Response: thank you for acknowledging that .we the authors tried to address included all the data 

5. Is the manuscript presented in an intelligible fashion and written in standard English?

Reviewer #1: Yes

Reviewer #2: Yes

Reviewer #3: Yes

Reviewer #4: Yes

Reviewer #5: No

Response: Thank you very much. We have edited it again using the free editing chance that is given to researcher until august 31 at Research Square is currently offering free use of an artificial intelligence editing tool during the Beta period called Automated Editing tool using the following links https://www.researchsquare.com/static/pdf/automated-editing-tool-instructions.pdf

6. Review Comments to the Author

Response: thanks .We tried to address each and every question/concern step by step

Reviewer #1: Thank you much improved

I can see you have put a lot of work in and i think it is useful now.

Response: thank you for acknowledging that dear our respected reviewer 

Still riddled with grammar and language issues.

I started pointing out and then got frustrated so below list is not complete. Needs proofreading and correctig again please.

Thank you very much. We have edited it again using the free editing chance that is given to researcher until august 31 at Research Square is currently offering free use of an artificial intelligence editing tool during the Beta period called Automated Editing tool using the following links https://www.researchsquare.com/static/pdf/automated-editing-tool-instructions.pdf.we have attached as a supplementary fields the edited

Methods

sample size first paragraph:

Since there is no any study in our countries - correct

Spelling collage - college

Since there is no any study in our countries

Thank you very much. We have corrected all

Reviewer #2: The grammatical errors have been largely addressed. In the recommendation section, the authors should improve to include other dissemination strategies that can be more effective in rural context such as the study sight. See attachment

Response: we the authors would like to acknowledge for your effort to improve our paper. We have got both the attached documents ad we tried to address per your suggestion. We thanks 

Reviewer #3: Dear authors,

You have made a great effort with this new article submission. You have included in your submission a marked-up copy of your manuscript highlighting changes and a rebuttal letter responding also to each point raised by the editor and reviewers. Both documents show clearly your new work based in all editor and reviewers’ previous observations. You have improved especially theoretical background and methodological aspects and analysis following our suggestions. The information regarding written informed consent for your participants and the explanation regarding ethical issues is now clear. The questionnaire in English is so helpful to understand more in depth the results. Following the suggestion of the editor and reviewers you have made extensive edition by a native speaker. The abstract has been improved following the reviewers’ suggestions. Authors have included the main objective of the article and now there is information regarding findings also in the abstract. The theoretical background has now new references giving the paper more consistency and comparativeness with other regional areas. Now you have used the latest report of data including different regions and you have modified the WHO strategies. In the Methods section there is new information answering many diverse comments from the reviewers. You have improved this section correctly, especially regarding participant recruitment method and demographic details. Tables about participants and the recruitment help in this way. The explanation on sample size and power calculation is also clear now. Information regarding how the variables were coded is now improved and the clarification about the multivariable regression instead of multivariate analysis is now clear. The information about the questionnaire is now improved and uploaded in the system. The discussion has now more in-depth analysis, including different finding around the world linked with your results.

Response: thank you for acknowledging that dear our respected reviewer. We hope we have tried to address the rest concerns as much as we can 

The conclusion and recommendation section needs more development. You have improved the conclusions, but it is to short and the recommendations are still so vague. You could start this section confirming (or not) if your main objective (included now in the abstract) has been reached with your research work. Later you could also comment if you have found some correlations or not in your work. Finally, you could provide more details in your recommendations.

Response: thank you for dear our respected reviewer. We hope we have tried to address the rest concerns as much as we can.

Reviewer #4: The authors have addressed my critiques regarding the presentation and analysis of the data. The revised presentation of the sample, variables and the analysis is clear and apropriate.

Response: thank you for acknowledging that dear our respected reviewer. We hope we have tried to address the rest concerns as much as we can 

Reviewer #5: The KAP Survey Model stands for Knowledge, Attitudes, and Practices. The title may reflect this order. Also it is more logical.

Response: dear reviewer thanks for your critical comments, we have modified it. See the track change 

The manuscript lacks a more convincing argumentation on why COVID19' KAP is important amongst students. In other words, why to study that population and not another one, or the general population, where poverty is expected to be more prevalent and interventions high priority.

Response: dear reviewer we thanks for your feedback. The rational of studying the students KAP is stated as follows: The pandemic also puts the whole educational system in to difficult situations; predominantly, undergraduate students represented a special group that was at the ages to get independence and freedom of life but with inadequate experiences. Consequently, their KAP were suggested to be significantly affected by the pandemic, which needed to be explored. The other thing is that we have assesse the KAP of the community after addressing this group.

In Methods (section entitled ‘Sample size determination and sampling procedure (procurement procedures’)

The authors wrote as new text ‘Since we have used 50% proportion, our sample can be considered as a representative of a larger population’. This sentence makes no sense and the authors should remove it from the manuscript. Setting a 50% level for each of the variables in the survey is helpful as it is a conservative approach when no previous knowledge is available on expected levels of a variable within a population. Setting that level ensures that the sample size will be sufficient for any level as a 50% prevalence or proportion generates the highest requirement for sample, keeping other known parameters constant.

Response: thank you dear our respected reviewer. We have removed it per your golden feedback 

Importantly, the authors mentioned several times (in manuscript and reply to reviewers) that they intended to represent the student population in that university. If that is so, then the described two stage sampling strategy selected by the authors is not self-weighted, and the authors need to calculate weights and incorporate those into their analysis. In addition there is attrition from the 'planned' sample to the 'collected' sample but little detail on circumstances (not present in dormitory at time of interview, refused to participate, provided incomplete answers or partial answers).

Response: Dear our kid reviewer thank you for your contractive concern .as you describe the Our study sample was obtained by a two-stage cluster sampling technique In the first stage, out of ten collages, four of them (collage of engendering, computing science, business and economics and collage of agriculture) were selected by simple random sampling, and the sample size was allocated proportionally to obtain the required sample size from each selected college. The sampling fraction is approximately equal to seven for all colleges. We also add 1.5 design effects. In the second stage, the study participants were selected from each year (see supplmetry file 1) and sections of the selected collage using a systematic random sampling technique every 7th student from each college registrar office log-book. After that, a dorm-to-dorm visit was used to obtain the sampled students. Students in each selected collage that were present during the data collection period and were able to give responses were included in the study. Those who had mental or physical disabilities and could not fill out the questionnaires (unwilling to participate in to the study) were excluded from the study. Frankly speaking we have proportionally allocated the sample size for each collage. But ,we didn’t code for each it like that .sorry please understand us dear reviewer

In the Discussion section, the authors wrote the following new text:

‘A supportive results were reported in China [13],Jordan [15] and India [12]. However, finding from Saudi Arabia 31.9%[16], Middle Eastern Arabic countries 17.0%[17],India 19.36% [18],in Tanzania [19] and in Uganda [20] showed that most of the community had inadequate Knowledge .This might be due to the late confirmed case report of COVID-19 in Ethiopia which might gave a time to know about the disease. Additionally, the devastating news reported about the disease, and the WHO declaration of the disease as a pandemic due to its sever pathogenicity and communicability [7, 8]. might also have increased the students’ knowledge.’ The first sentence above is not good English but the authors claimed the paper to be edited and polished by professionals. It does not seem to be the case here. Another example is below.‘This finding is in line with a study done in Indonesian[31], Nepalese[32], Sudan[30] and South Wollo Zone[33]’. Overall, the discussion is gaining some importance and becoming more balanced but requires better expression of the main ideas, and improving clarity. I see many punctuation errors.

Response: thank you very much .The comments of the reviewer were highly insightful and enabled us to greatly improve the quality of our manuscript. Thank you for that .Therefore, based on the reviewer concerns we have made extensive edition in our manuscript. We have edited using the free editing chance that is given to researcher until august 31 at Research Square is currently offering free use of an artificial intelligence editing tool during the Beta period called Automated Editing tool using the following links https://www.researchsquare.com/static/pdf/automated-editing-tool-instructions.pdf . We have attached it as a supplementary file. See it 

References no. 1, 10 and 11, seem incomplete.

Response: Dear our respected reviewers, we would like to thanks once again for your critical review of the paper. We have cited with the full citation (we have removed the www.way of citation we cited it again. See it 

7. PLOS authors have the option to publish the peer review history of their article (what does this mean?). If published, this will include your full peer review and any attached files.

Do you want your identity to be public for this peer review? For information about this choice, including consent withdrawal, please see our Privacy Policy.

Reviewer #1: Yes: PW Hodkinson

Reviewer #2: Yes: Dr John Bosco Azigwe

Reviewer #3: Yes: Ramon Flecha

Reviewer #4: No

Reviewer #5: Yes: Jose Manuel Rodriguez-Llanes

Response: Dear our respected reviewers, we would like to thanks for your effort of giving the constructive feedback throughout the paper progresses. Without your feedback the output of this finding may loss its strength. For that thank you

Stay safe

Thanks

---

## [Decision Letter · Decision Letter 2]

30 Sep 2020

PONE-D-20-14158R2

Assessment of undergraduate student knowledge, attitude, and practices, towards COVID-19 in Debre Berhan University, Ethiopia

PLOS ONE

Dear Dr. Aynalem,

Thank you for submitting your manuscript to PLOS ONE. After careful consideration, we feel that it has merit but does not fully meet PLOS ONE’s publication criteria as it currently stands. Therefore, we invite you to submit a revised version of the manuscript that addresses the points raised during the review process.

The reviewers have emphasised once more 3 different areas where the manuscript does require improvement and clarifications:

- Presentation and overall writing quality of the paper, with references still incomplete, punctuation errors etc - please assume the paper is going to be read widely, and by many different readers which may include policymakers, government officials, other colleagues and patients/citizens, so I would encourage care on the overall appearance.

- Further details so that readers understand how the authors made their design selfweighted. Bit more clarity is needed. The argumentation for a 50% proportion is not obvious for a non expert reader and should be better supported with an explanatory reference.

- Review again the Discussion and recommendations section, because they remain short in length and not directly connected with the main objective included in the abstract.

We look forward to receiving your revised manuscript.

Kind regards,

Enrique Castro-Sánchez

Academic Editor

PLOS ONE

Journal Requirements:

1) We note that Table 2 appears to be mislabeled as a second Table 1. Please revise the table name for clarity.

Reviewers' comments:

Reviewer's Responses to Questions

**Comments to the Author**

1. If the authors have adequately addressed your comments raised in a previous round of review and you feel that this manuscript is now acceptable for publication, you may indicate that here to bypass the “Comments to the Author” section, enter your conflict of interest statement in the “Confidential to Editor” section, and submit your "Accept" recommendation.

Reviewer #2: All comments have been addressed

Reviewer #3: (No Response)

Reviewer #5: (No Response)

2. Is the manuscript technically sound, and do the data support the conclusions?

Reviewer #2: Yes

Reviewer #3: Yes

Reviewer #5: Partly

3. Has the statistical analysis been performed appropriately and rigorously? 

Reviewer #2: Yes

Reviewer #3: Yes

Reviewer #5: I Don't Know

4. Have the authors made all data underlying the findings in their manuscript fully available?

Reviewer #2: Yes

Reviewer #3: Yes

Reviewer #5: No

5. Is the manuscript presented in an intelligible fashion and written in standard English?

Reviewer #2: Yes

Reviewer #3: Yes

Reviewer #5: Yes

6. Review Comments to the Author

Reviewer #2: The authors in their recommendations should be more specific on community based strategies for information dissemination on covid19

Reviewer #3: Dear authors,

I have seen the new version of your article (R2) with track changes and without it. You have made extensive edition in your manuscript using the free use of an artificial intelligence editing tool. The new extensive edited manuscript by a professional language editor is so valuable. You have worked hard in the final proposal in this third stage. I suggest to review again the Discussion and recommendations section, because they remain short in lenght and not directly connected with your main objective included in the abstract.

Reviewer #5: Thanks for the opportunity to revise this MS.

The references are yet incomplete. I can still see punctuation errors in the MS and the authors need to provide further details so that readers understand how they made their design selfweighted. Bit more clarity is needed. The argumentation for a 50% proportion is not obvious for a non expert reader and should be better supported with a explanatory reference.

7. PLOS authors have the option to publish the peer review history of their article (what does this mean?). If published, this will include your full peer review and any attached files.

Reviewer #2: **Yes: **Dr John Bosco Azigwe

Reviewer #3: **Yes: **Ramon Flecha

Reviewer #5: **Yes: **Jose Manuel Rodriguez-Llanes

---

## [Author Response · Author response to Decision Letter 2]

16 Oct 2020

Dear Academic Editor!

Plose one 

Response to Academic Editor 

I am pleased to resubmit(R3) for publication version of “Assessment of undergraduate student knowledge, practices, and attitude towards COVID-19 in Debre Berhan University, Ethiopia for a review as original research in Plose one 

The comments of the editor and the reviewers were highly insightful and enabled us to greatly improve the quality of our manuscript. Therefore, based on the editor’s and the reviewers’ concerns we have made extensive edition in our manuscript. The comments of the editor and the reviewers were highly insightful and enabled us to greatly improve the quality of our manuscript. Therefore, based on the editor’s and the reviewers’ concerns we have made extensive edition in our manuscript. Especially we have extensively edited the manuscript again by a professional language editor, at Excision Editing (a fluent, native Australian, English-language speaker thoroughly edited the manuscript for language usage, spelling, and grammar) before submitting the revised version. The formatting of the text and document (text sizes and grammatical errors) were also edited. His name is called Dr. Ryan Bell(CEO and Chief Editor Excision Editing)

 In the following pages, we have addressed yours’ concerns in a point by point format. 

 We look forward to hearing from you at your earliest convenience. 

Thank you for your consideration of this manuscript! 

Kind regards,

Yared Asmare Aynalem.

On behalf of authors

PONE-D-20-14158R2

Assessment of undergraduate student knowledge, attitude, and practices, towards COVID-19 in Debre Berhan University, Ethiopia

PLOS ONE

Dear Dr. Aynalem,

Thank you for submitting your manuscript to PLOS ONE. After careful consideration, we feel that it has merit but does not fully meet PLOS ONE’s publication criteria as it currently stands. Therefore, we invite you to submit a revised version of the manuscript that addresses the points raised during the review process.

Response: Thank you very much for allowing us to revise our manuscript again. We have tried to response each concerns as much as possible. We have submitted a revised version of the manuscript.

The reviewers have emphasised once more 3 different areas where the manuscript does require improvement and clarifications:

- Presentation and overall writing quality of the paper, with references still incomplete, punctuation errors etc - please assume the paper is going to be read widely, and by many different readers which may include policymakers, government officials, other colleagues and patients/citizens, so I would encourage care on the overall appearance.

Response: Thank you very much. We the authors tried to edit the whole manuscript per the reviewer and the editor suggestion 

- Further details so that readers understand how the authors made their design selfweighted. Bit more clarity is needed. The argumentation for a 50% proportion is not obvious for a non expert reader and should be better supported with an explanatory reference.

Respose Dear our respected review ereditor , thanks for spending your time to this extent. Which helps to improve our paper as your concern, we the authors tried to make the reference as completed as possible. Additionally, we have made an extensive edition of the paper including the punctuation issue. We also support the argumentation of using a 50% proportion with citation. For more please also see the color change

- Review again the Discussion and recommendations section, because they remain short in length and not directly connected with the main objective included in the abstract.

Response: Thank you very much. We have rewrite the Discussion and recommendations section as detailed ad possible 

Response: Thank you. We have included ‘Response to Reviewers', 'Revised Manuscript with Track Changes Manuscript separately 

Response: Thank you for the concern. There is no any change 

We look forward to receiving your revised manuscript.

Kind regards,

Enrique Castro-Sánchez

Academic Editor

Response: Thank you for the concern. Not applicable

PLOS ONE

Journal Requirements:

1) We note that Table 2 appears to be mislabeled as a second Table 1. Please revise the table name for clarity.

Response: Thank you. We have labeled it. See the color change 

Reviewers' comments:

Reviewer's Responses to Questions

Comments to the Author

1. If the authors have adequately addressed your comments raised in a previous round of review and you feel that this manuscript is now acceptable for publication, you may indicate that here to bypass the “Comments to the Author” section, enter your conflict of interest statement in the “Confidential to Editor” section, and submit your "Accept" recommendation.

Reviewer #2: All comments have been addressed

Reviewer #3: (No Response)

Reviewer #5: (No Response)

Response: thank you for acknowledging that 

2. Is the manuscript technically sound, and do the data support the conclusions?

Reviewer #2: Yes

Reviewer #3: Yes

Reviewer #5: Partly

Response: thank you for acknowledging that .we the authors tried to address it again 

3. Has the statistical analysis been performed appropriately and rigorously?

Reviewer #2: Yes

Reviewer #3: Yes

Reviewer #5: I Don't Know

Response: thank you for acknowledging that . 

4. Have the authors made all data underlying the findings in their manuscript fully available?

Reviewer #2: Yes

Reviewer #3: Yes

Reviewer #5: No

Response: thank you for acknowledging that .we the authors tried to address included all the data 

5. Is the manuscript presented in an intelligible fashion and written in standard English?

Reviewer #2: Yes

Reviewer #3: Yes

Reviewer #5: Yes

Response: thank you for acknowledging that

6. Review Comments to the Author

Response: thank you .We tried to answer the reviewers concern step by step

Reviewer #2: The authors in their recommendations should be more specific on community based strategies for information dissemination on covid19

Response: dear editor very thank you for your golden feedback. We the authors tried to make it as specific as possible. Please see the color change 

Reviewer #3: Dear authors,

I have seen the new version of your article (R2) with track changes and without it. You have made extensive edition in your manuscript using the free use of an artificial intelligence editing tool. The new extensive edited manuscript by a professional language editor is so valuable. You have worked hard in the final proposal in this third stage. I suggest to review again the Discussion and recommendations section, because they remain short in lenght and not directly connected with your main objective included in the abstract.

Response: thank you for acknowledging that .we the authors tried to address the issue .we reviewed again the Discussion and recommendations section, to make it as long as possible and to make it connected with our main objective included in the abstract. see the color.

Reviewer #5: Thanks for the opportunity to revise this MS.

The references are yet incomplete. I can still see punctuation errors in the MS and the authors need to provide further details so that readers understand how they made their design selfweighted. Bit more clarity is needed. The argumentation for a 50% proportion is not obvious for a non expert reader and should be better supported with a explanatory reference.

Response: Dear our respected reviewer, we are glad to have a reviewer like you .thanks for spending your time to this extent. Which helps to improve our paper.as your concern, we the authors tried to make the reference as completed as possible. additionally, we have made an extensive edition of the paper including the punctuation issue. We also support the argumentation of using a 50% proportion with citation. For more please also see the color change 

7. PLOS authors have the option to publish the peer review history of their article (what does this mean?). If published, this will include your full peer review and any attached files.

Response: thank you .Yes 

Do you want your identity to be public for this peer review? For information about this choice, including consent withdrawal, please see our Privacy Policy.

Reviewer #2: Yes: Dr John Bosco Azigwe

Reviewer #3: Yes: Ramon Flecha

Reviewer #5: Yes: Jose Manuel Rodriguez-Llanes

Response: thank you for your volunteer of your identity to be public for this peer review

---

## [Decision Letter · Decision Letter 3]

10 Dec 2020

PONE-D-20-14158R3

Assessment of undergraduate student knowledge, attitude, and practices, towards COVID-19 in Debre Berhan University, Ethiopia

PLOS ONE

Dear Dr. Aynalem,

Thank you for submitting your manuscript to PLOS ONE. After careful consideration, we feel that it has merit but does not fully meet PLOS ONE’s publication criteria as it currently stands. Therefore, we invite you to submit a revised version of the manuscript that addresses the points raised during the review process.

I thank you for the revisions that you have conducted, but I would urge you once more to carefully review the manuscript in terms of clarity and grammar. This is not the usual request for authors to ensure that the English language in the manuscript is adequate, but to ensure that the manuscript is readable and understandable. The paper has had several rounds of reviews and opportunities for these issues to be resolved, so I would be grateful if you could ensure they are resolved. It is disconcerting to find so many issues despite assurances that they have been resolved.

In addition to the comments and recommendations mentioned by the reviewers, I have read the manuscript again and I would like the authors to resolve these pending matters:

**Background**

2nd para, why is the mortality compared with the flu?

End of 2nd para, and start of 3rd para, the data does not match- please review and clarify.

**Ethics statement**

Review wording- "ethical approval commute" should be committee- please review across the manuscript.

Review the paragraph for clarity.

Please clarify authors contribution, currently 2nd and 6th authors, and 3rd, 4th, 5th are mentioned as contributing equally.

**Background**

2nd para, says "2215 countries", must be a typo.

Reference in pg 16 [9,10] appears in cursive font, please edit.

End of page: "study conducted in different parts of there world" - say "international" or say which parts of the world.

**Methods**

Please remove "Abve sea level and temp" unless relevant to the study.

Please edit "Collage of ..." To "College of..."

Justify sociodemographic variables, marital status, religion - why collected?

Sociodemographic appears in study variables and data collection tool.

Was the data collection validated?

"Additional data were collected through..." - what additional data?

Was tool validated in students or only faculty?

"Ensure genuine replies..." - how did they do that?

**Ethics consideration**

Commute = committee

Paper says participation anonymous but later on it mentioned that collects personal details like phone? Please clarify

**Results**

What was the total number of participants approached, to give the 86.1% rate?

**Discussion**

End of 1st para: please add evidence that "Students get regular update through the medias"

2nd para: what educational program? How can severity of disease lead to increased knowledge? Any evidence

2nd and 3rd para need punctuation mark at the end.

We look forward to receiving your revised manuscript.

Kind regards,

Enrique Castro-Sánchez

Academic Editor

PLOS ONE

Reviewers' comments:

Reviewer's Responses to Questions

**Comments to the Author**

1. If the authors have adequately addressed your comments raised in a previous round of review and you feel that this manuscript is now acceptable for publication, you may indicate that here to bypass the “Comments to the Author” section, enter your conflict of interest statement in the “Confidential to Editor” section, and submit your "Accept" recommendation.

Reviewer #3: All comments have been addressed

2. Is the manuscript technically sound, and do the data support the conclusions?

Reviewer #3: Yes

3. Has the statistical analysis been performed appropriately and rigorously? 

Reviewer #3: Yes

4. Have the authors made all data underlying the findings in their manuscript fully available?

Reviewer #3: Yes

5. Is the manuscript presented in an intelligible fashion and written in standard English?

Reviewer #3: Yes

6. Review Comments to the Author

Reviewer #3: Authors have adequately addressed my comments in this new version of the article. Discussion section provides more concrete information.

7. PLOS authors have the option to publish the peer review history of their article (what does this mean?). If published, this will include your full peer review and any attached files.

Reviewer #3: **Yes: **Ramon Flecha

---

## [Author Response · Author response to Decision Letter 3]

10 Jan 2021

Dear Academic Editor!

PLOSE ONE 

Response to Academic Editor 

I am pleased to resubmit revision (R4) for publication version of “Assessment of undergraduate student knowledge, practices, and attitude towards COVID-19 in Debre Berhan University, Ethiopia for a review as original research in Plose one .The comments of the editor were highly insightful and enabled us to greatly improve the quality of our manuscript. Therefore, based on the editor’s concerns we have made extensive edition in our manuscript. Especially we have extensively edited the manuscript again for language usage, spelling, and grammar before submitting the revised version. The formatting of the text and document (text sizes and grammatical errors) were also edited. In the following pages, we have addressed yours’ concerns in a point by point format. 

 We look forward to hearing from you at your earliest convenience. 

Thank you for your consideration of this manuscript! 

Kind regards,

Yared Asmare Aynalem.

On behalf of authors

PONE-D-20-14158R3

Assessment of undergraduate student knowledge, attitude, and practices, towards COVID-19 in Debre Berhan University, Ethiopia

PLOS ONE

Dear Dr. Aynalem,

Thank you for submitting your manuscript to PLOS ONE. After careful consideration, we feel that it has merit but does not fully meet PLOS ONE’s publication criteria as it currently stands. Therefore, we invite you to submit a revised version of the manuscript that addresses the points raised during the review process.

Response: Thank you very much for allowing us to revise our manuscript again. We have tried to response each concerns as much as possible. We have submitted a revised version of the manuscript

I thank you for the revisions that you have conducted, but I would urge you once more to carefully review the manuscript in terms of clarity and grammar. This is not the usual request for authors to ensure that the English language in the manuscript is adequate, but to ensure that the manuscript is readable and understandable. The paper has had several rounds of reviews and opportunities for these issues to be resolved, so I would be grateful if you could ensure they are resolved. It is disconcerting to find so many issues despite assurances that they have been resolved.

Response: Thank you for allowing us to edit it again. The comments of the editor and the reviewers were highly insightful and enabled us to greatly improve the quality of our manuscript. Therefore, based on your concerns we have made extensive edition in our manuscript. Your comments were highly insightful and enabled us to greatly improve the quality of our manuscript. Therefore, based on the editor’s concerns we have made extensive edition in our manuscript. Especially we have extensively edited the manuscript again for language clarity thoroughly edited the manuscript for language usage, spelling, and grammar before submitting the revised version. The formatting of the text and document (text sizes and grammatical errors) were also edited. 

In addition to the comments and recommendations mentioned by the reviewers, I have read the manuscript again and I would like the authors to resolve these pending matters:

Response: Dear our respected editor thank you very much for raising the following issues 

Background

2nd para, why is the mortality compared with the flu?

Response: Dear our respected editor thanks you very much. The reason for comparing COVID-19 with the flu is that to show the proportion of mortality as a new outbreak. At the time of seasonal flue outbreak the mortality rate were almost 1%.more over covid mortality rate is almost more than 7 percent .This death rate is particularly alarming. If you are still not comfortable with this sentence, we will remove it.

End of 2nd para, and start of 3rd para, the data does not match- please review and clarify.

Response: Thanks you very much our respected editor .We have rewrite it again; please see the color change from the main document 

Ethics statement

Review wording- "ethical approval commute" should be committee - please review across the manuscript. Review the paragraph for clarity.

Response: Thanks you very much our respected editor. We have copy edited it. see also the color change from the ma document 

Response: Thank you dear editor. We have clearly descried the contribution on the on the online author contribution form.

Background

2nd para, says "2215 countries", must be a typo.

Response: thanks our beloved editor We have rewritten it. Please See the color change 

Reference in pg 16 [9,10] appears in cursive font, please edit.

Response: Thanks .We has corrected it. Please see the color change 

End of page: "study conducted in different parts of their world" - say "international" or say which parts of the world.

Response: Thanks .We tried to rephrase replace with your suggestive appropriate word so called "international"

Methods

Please remove "Abve sea level and temp" unless relevant to the study. Please edit "Collage of ..." To "College of..."

Response: Thanks .We has corrected it. See the color change 

Justify socio-demographic variables, marital status, religion - why collected?

Response: we would like to thanks our editor for raising these important concerns. The rational for collecting socio-demographic variables, marital status, and religion is as follows

1. In our country context, the distribution of COVID_19 is high among aged individual most of adult generation perceived that they are less risky of contracting with it. For that they didn’t want to get even information on it. Therefore, we have collected it as a factor.

2. Most of the community including students talk as COVID is a sin of us.it is from alah,God. So that, not practicing a percussion. This document and risk assessment tool provides practical guidance and recommendations to support the special role of religious leaders, faith-based organizations, and faith communities in COVID-19 education, preparedness, and response.

3. Additionally, differ study conducted globally, shows that socio-demographic variables, marital status, religion are linked with COVID-19 KAP. Here, we examine how KAP varies across fundamental socio-demographic characteristics, including age, sex, civil status, individual disposable income, religion of. We have collected it to for identifying the gap to intervene on it .

Was the data collection validated?

Was tool validated in students or only faculty?

"Ensure genuine replies..." - how did they do that?

Thanks dear reviewer..!!! The proposed tool was developed and validated by a multidisciplinary working group of infectious disease physicians, lecturers (Public health, nurse environmental health professionals, and infectious diseases Public health professionals. The working group regularly reviewed literature to select important characteristics and outcomes for inclusion. We also used a 

The development phase was conducted at Debre birhan University health science collage and the pilot validation phase was conducted at the nursing department students. We have also done a pretest from 5 % of the student. The team members formed a working group that met via telephone or video conferences at least biweekly 

Ethics consideration

Commute = committee,Paper says participation anonymous but later on it mentioned that collects personal details like phone? Please clarify

Response: Thanks .We has corrected it. See the color change 

Results

What was the total number of participants approached, to give the 86.1% rate? 634

Response: Thanks dear editor.as we have stated in sample size determination, we have included approached to a total of 634 participate. 

Discussion

End of 1st para: please add evidence that "Students get regular update through the medias"

Response: Dear editor thanks for raising the concern. We have given a possible justifications based o our experience of giving the student about the outbreak COVID-19. We have given to them a health education program for all faculty of the university before the school was closed. Possibly this may increase their knowledge 

2nd and 3rd para need punctuation mark at the end. 

Response : Dear editor thanks for raising the concern. We have tried to edit all parts of the paper.

Response: thank you for your critical review dear editor. We have corrected it .See also the color change 

 Response: Thank you. We have included ‘Response to Reviewers', 'Revised Manuscript with Track Changes Manuscript separately 

editorialmanager.com/pone/)

Response: Thank you. We dot like to change the financial disclosure

We look forward to receiving your revised manuscript.

Kind regards,

Enrique Castro-Sánchez

Academic Editor

PLOS ONE

Reviewers' comments:

Reviewer's Responses to Questions

Comments to the Author

1. If the authors have adequately addressed your comments raised in a previous round of review and you feel that this manuscript is now acceptable for publication, you may indicate that here to bypass the “Comments to the Author” section, enter your conflict of interest statement in the “Confidential to Editor” section, and submit your "Accept" recommendation.

Reviewer #3: All comments have been addressed

Response: dear reviewer, we thank you for spending your valuable time to improve this paper. We have a grate appreciation for helping us 

2. Is the manuscript technically sound, and do the data support the conclusions?

Reviewer #3: Yes

Response: Thank you dear reviewer

3. Has the statistical analysis been performed appropriately and rigorously?

Reviewer #3: Yes

Response: Thank you dear reviewer

4. Have the authors made all data underlying the findings in their manuscript fully available?

Reviewer #3: Yes

Response: Thank you dear reviewer

5. Is the manuscript presented in an intelligible fashion and written in standard English?

Reviewer #3: Yes

 Response: Thank you dear reviewer

6. Review Comments to the Author

Reviewer #3: Authors have adequately addressed my comments in this new version of the article. Discussion section provides more concrete information.

Response: Thank you dear reviewer. It is your effort too

7. PLOS authors have the option to publish the peer review history of their article (what does this mean?). If published, this will include your full peer review and any attached files.

Do you want your identity to be public for this peer review? For information about this choice, including consent withdrawal, please see our Privacy Policy.

Reviewer #3: Yes: Ramon Flecha

Response: Thank you dear reviewer

Dear all (both the reviewer ad editors),we would like to appreciate all your effort for your effort to improve our paper .may God paid all your effort. We have get differ research skills from your point of view

Thanks

---

## [Editor Report · Decision Letter 4]

19 Jan 2021

PONE-D-20-14158R4

Assessment of undergraduate student knowledge, attitude, and practices, towards COVID-19 in Debre Berhan University, Ethiopia

PLOS ONE

Dear Dr. Aynalem,

Thank you for submitting your manuscript to PLOS ONE. After careful consideration, we feel that it has merit but once again, it does not fully meet PLOS ONE’s publication criteria as it currently stands. Therefore, we invite you to submit a revised version of the manuscript that addresses the points raised during the review process.

I have reviewed again, and annotated, the revised version of the manuscript. As you can see, several language and copyeditting issues remain, which I have highlighted. The reason why I am emphasising this aspect of the process is that PLOS ONE manuscripts are published as submitted by the authors, as per the accepted version. 

Additionally, some of the responses provided to justify some of the concerns and queries raised are not adequate as they really do not offer a justification.

2nd para, why is the mortality compared with the flu? Response: Dear our respected editor thanks you very much. The reason for comparing COVID-19 with the flu is that to show the proportion of mortality as a new outbreak. At the time of seasonal flue outbreak the mortality rate were almost 1%.more over covid mortality rate is almost more than 7 percent .This death rate is particularly alarming. If you are still not comfortable with this sentence, we will remove it. 

** Please remove or simply state that the mortality of covid is estimated at more than 7%**

Background 

2nd para, says "2215 countries", must be a typo. Response: thanks our beloved editor We have rewritten it. Please See the color change 

**The text still says 2215 countries. Please review once more, do you mean 215 countries?**

Justify socio-demographic variables, marital status, religion - why collected? 

Response: we would like to thanks our editor for raising these important concerns. The rational for collecting socio-demographic variables, marital status, and religion is as follows 

1. In our country context, the distribution of COVID_19 is high among aged individual most of adult generation perceived that they are less risky of contracting with it. For that they didn’t want to get even information on it. Therefore, we have collected it as a factor. 

Most of the community including students talk as COVID is a sin of us.it is from alah,God. So that, not practicing a percussion. This document and risk assessment tool provides practical guidance and recommendations to support the special role of religious leaders, faith-based organizations, and faith communities in COVID-19 education, preparedness, and response. **If that is the case, and there is evidence of students talking about covid19 as a sin, then this argument needs to be included in the paper.**Additionally, differ study conducted globally, shows that socio-demographic variables, marital status, religion are linked with COVID-19 KAP. **Can you please include references to those studies that indicate that marital status and religion are linked with COVID-19 KAP?**

Was the data collection validated?

Was tool validated in students or only faculty? "Ensure genuine replies..." - how did they do that? 

**The authors did not respond to the query - how did the interviewers ensured that genuine replies were give?**

Ethics consideration 

Commute = committee,Paper says participation anonymous but later on it mentioned that collects personal details like phone? Please clarify 

Response: Thanks .We has corrected it. See the color change 

** There is still a mention to the collection of phone number. The authors did not respond to the query. Why was the phone number of the participants collected?**

Results 

What was the total number of participants approached, to give the 86.1% rate? 634 

Response: Thanks dear editor.as we have stated in sample size determination, we have included approached to a total of 634 participate. 

**The 634 number is not included in the manuscript, can you include?**

Discussion 

End of 1st para: please add evidence that "Students get regular update through the medias" 

Response: Dear editor thanks for raising the concern. We have given a possible justifications based o our experience of giving the student about the outbreak COVID-19. We have given to them a health education program for all faculty of the university before the school was closed. Possibly this may increase their knowledge 

**In the Discussion, authors are allowed, and encouraged, to hypothesise about the reasons explaining their findings. However, the authors also need to make sure that the reader knows whether there is any data supporting such explanations. IF you say that "in your experience students get regular update through media", then you should include that in the paper.**

We look forward to receiving your revised manuscript.

Kind regards,

Enrique Castro-Sánchez

Academic Editor

PLOS ONE

---

## [Author Response · Author response to Decision Letter 4]

30 Jan 2021

Dear Academic Editor!

Plose one 

Response to Academic Editor 

I am pleased to resubmit(R5) for publication version of “Assessment of undergraduate student knowledge, practices, and attitude towards COVID-19 in Debre Berhan University, Ethiopia for a review as original research in Plose one 

The comments of the editor were highly insightful and enabled us to greatly improve the quality of our manuscript. Therefore, based on the editor’s concerns we have made extensive edition in our manuscript. The comments of the editor and were highly insightful and enabled us to greatly improve the quality of our manuscript. Therefore, based on the editor’s and the reviewers’ concerns we have made extensive edition in our manuscript. Especially we have extensively edited the manuscript again per editors feedback thoroughly edited the manuscript for language usage, spelling, and grammar) before submitting the revised version. The formatting of the text and document (text sizes and grammatical errors) were also edited. In the following pages, we have addressed yours’ concerns in a point by point format. 

 We look forward to hearing from you at your earliest convenience. 

Thank you for your consideration of this manuscript! 

Kind regards,

Yared Asmare Aynalem.

On behalf of authors

PONE-D-20-14158R4

Assessment of undergraduate student knowledge, attitude, and practices, towards COVID-19 in Debre Berhan University, Ethiopia

PLOS ONE

Dear Dr. Aynalem,

Thank you for submitting your manuscript to PLOS ONE. After careful consideration, we feel that it has merit but once again, it does not fully meet PLOS ONE’s publication criteria as it currently stands. Therefore, we invite you to submit a revised version of the manuscript that addresses the points raised during the review process.

I have reviewed again, and annotated, the revised version of the manuscript. As you can see, several language and copyediting issues remain, which I have highlighted. The reason why I am emphasizing this aspect of the process is that PLOS ONE manuscripts are published as submitted by the authors, as per the accepted version. 

Additionally, some of the responses provided to justify some of the concerns and queries raised are not adequate as they really do not offer a justification.

2nd para, why is the mortality compared with the flu? Response: Dear our respected editor thanks you very much. The reason for comparing COVID-19 with the flu is that to show the proportion of mortality as a new outbreak. At the time of seasonal flue outbreak the mortality rate were almost 1%.more over covid mortality rate is almost more than 7 percent .This death rate is particularly alarming. If you are still not comfortable with this sentence, we will remove it.

 Please remove or simply state that the mortality of covid is estimated at more than 7%

Response: thanks dear editor. We have modified it. See the track change 

Background

2nd para, says "2215 countries", must be a typo. Response: thanks our beloved editor we have rewritten it. Please see the color change

>The text still says 2215 countries. Please review once more, do you mean 215 countries?

Response: our respected editor, a great apologize to the inconvenience 

It’s to say two hundred fifteen countries

Justify socio-demographic variables, marital status, religion - why collected?

Response: we would like to thanks our editor for raising these important concerns. The rational for collecting socio-demographic variables, marital status, and religion is as follows

1. In our country context, the distribution of COVID_19 is high among aged individual most of adult generation perceived that they are less risky of contracting with it. For that they didn’t want to get even information on it. Therefore, we have collected it as a factor.

2. Most of the community including students talk as COVID is a sin of us.it is from alah,God. So that, not practicing a percussion. This document and risk assessment tool provides practical guidance and recommendations to support the special role of religious leaders, faith-based organizations, and faith communities in COVID-19 education, preparedness, and response. If that is the case, and there is evidence of students talking about covid19 as a sin, then this argument needs to be included in the paper.

3. Additionally, differ study conducted globally, shows that socio-demographic variables, marital status, religion are linked with COVID-19 KAP. Can you please include references to those studies that indicate that marital status and religion are linked with COVID-19 KAP?

Response –dear editor once again we would like to acknowledge for your deep understanding of our paper. Even if we included the above mentioned variables, we dint got a significant association with our regression result. We have described them only them.so that we didn’t cited them. If you consider irrelevant, we may remove the variables .more of the above description that we have put were a rumor with in the community, that is why we want to see them

Thank you 

Ethics consideration

Commute = committee, Paper says participation anonymous but later on it mentioned that collects personal details like phone? Please clarify

Response: Thanks .We has corrected it. See the color change

 There is still a mention to the collection of phone number. The authors did not respond to the query. Why was the phone number of the participants collected?

Response : Dear editor thank you for the query. We have use the phone number of the student to trace them .cuse they were isolated to their dormitory 

Results

What was the total number of participants approached, to give the 86.1% rate? 634

Response: Thanks dear editor.as we have stated in sample size determination, we have included approached to a total of 634 participate.

>The 634 number is not included in the manuscript, can you include?

Response: Thank you. We have included it.

Discussion

End of 1st para: please add evidence that "Students get regular update through the Medias"

Response: Dear editor thanks for raising the concern. We have given a possible justifications based o our experience of giving the student about the outbreak COVID-19. We have given to them a health education program for all faculty of the university before the school was closed. Possibly this may increase their knowledge

>In the Discussion, authors are allowed, and encouraged, to hypothesise about the reasons explaining their findings. However, the authors also need to make sure that the reader knows whether there is any data supporting such explanations. IF you say that "in your experience students get regular update through media", then you should include that in the paper.

Response: dear reviewer we tried to cite all justification as per your valuable feedback 

We look forward to receiving your revised manuscript.

Kind regards,

Enrique Castro-Sánchez

Academic Editor

PLOS ONE

Response: Thank you. We have included ‘Response to Reviewers', 'Revised Manuscript with Track Changes Manuscript separately

---

## [Editor Report · Decision Letter 5]

16 Mar 2021

PONE-D-20-14158R5

Assessment of undergraduate student knowledge, attitude, and practices, towards COVID-19 in Debre Berhan University, Ethiopia

PLOS ONE

Dear Dr. Aynalem,

Thank you for submitting your manuscript to PLOS ONE. After careful consideration, we feel that it has merit but does not fully meet PLOS ONE’s publication criteria as it currently stands. Therefore, we invite you to submit a revised version of the manuscript that addresses the points raised during the review process.

*The following is an odd sentence and needs to be revised in the discussion: "This might be because of easily accessible to most students at home and everywhere through the mobile internet".*

We look forward to receiving your revised manuscript.

Kind regards,

Frank T. Spradley

Academic Editor

PLOS ONE
---

## [Author Response · Author response to Decision Letter 5]

19 Mar 2021

Dear Academic Editor!

Plose one 

Response to Academic Editor 

I am pleased to resubmit(R6) for publication version of “Assessment of undergraduate student knowledge, practices, and attitude towards COVID-19 in Debre Berhan University, Ethiopia for a review as original research in Plose one .The comments of the editor were highly insightful and enabled us to greatly improve the quality of our manuscript. Therefore, based on the editor’s concerns we have made extensive edition in our manuscript. The comments of the editor were highly insightful and enabled us to greatly improve the quality of our manuscript. Therefore, based on the editor’s concerns we have made extensive edition in our manuscript particularly the reference part. The formatting of the text and document (text sizes and grammatical errors) were also edited. In the following pages, we have addressed yours’ concerns in a point by point format. 

 We look forward to hearing from you at your earliest convenience. 

Thank you for your consideration of this manuscript! 

Kind regards,

Yared Asmare Aynalem.

On behalf of authors

PONE-D-20-14158R5

Assessment of undergraduate student knowledge, attitude, and practices, towards COVID-19 in Debre Berhan University, Ethiopia

PLOS ONE

Dear Dr. Aynalem,

Thank you for submitting your manuscript to PLOS ONE. After careful consideration, we feel that it has merit but does not fully meet PLOS ONE’s publication criteria as it currently stands. Therefore, we invite you to submit a revised version of the manuscript that addresses the points raised during the review process.

Response: Dear our respected editor thanks you very much for considerations. I have submitted a revised version of the manuscript that addresses the points raised during the review process as much as we can.

The following is an odd sentence and needs to be revised in the discussion: "This might be because of easily accessible to most students at home and everywhere through the mobile internet".

Response: Dear our respected editor thanks you very much for your critical review. We have removed this odd sentence

 Response: Thank you. We have included ‘Response to Reviewers', 'Revised Manuscript with Track Changes ad Manuscript separately 

Response: thanks for allowing us to update the financial disclosure. However, there is no any change. 

Response: thank you for recommending depositing the laboratory protocols in protocols.io. But it is not applicable

We look forward to receiving your revised manuscript.

Kind regards,

Frank T. Spradley

Academic Editor

Response: Thank you dear editor .we has set it 

PLOS ONE

Journal Requirements:

Please review your reference list to ensure that it is complete and correct. If you have cited papers that have been retracted, please include the rationale for doing so in the manuscript text, or remove these references and replace them with relevant current references. Any changes to the reference list should be mentioned in the rebuttal letter that accompanies your revised manuscript. If you need to cite a retracted article, indicate the article’s retracted status in the References list and also include a citation and full reference for the retraction notice

Response: thank you for the feedback dear our editor.as far as we searched we didn’t cite papers that have been retracted and no any changes to the reference list. We tried to search for papers that might retract from journal. However as far as we searched, we didn’t get it. if any please suggest us we will remove and replace it .We also tried to make the reference full as the journal guide line. See the track charge 

Thank you 

Keep safe

---

## [Editor Report · Decision Letter 6]

23 Mar 2021

PONE-D-20-14158R6

Assessment of undergraduate student knowledge, attitude, and practices, towards COVID-19 in Debre Berhan University, Ethiopia

PLOS ONE

Dear Dr. Aynalem,

Thank you for submitting your manuscript to PLOS ONE. After careful consideration, we feel that it has merit but does not fully meet PLOS ONE’s publication criteria as it currently stands. Therefore, we invite you to submit a revised version of the manuscript that addresses the points raised during the review process.

ACADEMIC EDITOR COMMENT: I did not request that the sentence I questioned earlier be removed. It is important for one of the previous revisions. Please REVISE that sentence to proper English and make it understandable and resubmit this manuscript. 

We look forward to receiving your revised manuscript.

Kind regards,

Frank T. Spradley

Academic Editor

PLOS ONE
---

## [Author Response · Author response to Decision Letter 6]

26 Mar 2021

Dear Academic Editor!

Plose one 

Response to Academic Editor 

I am pleased to resubmit(R6) for publication version of “Assessment of undergraduate student knowledge, practices, and attitude towards COVID-19 in Debre Berhan University, Ethiopia for a review as original research in Plose one .The comments of the editor were highly insightful and enabled us to greatly improve the quality of our manuscript. Therefore, based on the editor’s concerns we have made extensive edition in our manuscript. The comments of the editor were highly insightful and enabled us to greatly improve the quality of our manuscript. Therefore, based on the editor’s concerns we have made extensive edition in our manuscript particularly the reference part. The formatting of the text and document (text sizes and grammatical errors) were also edited. In the following pages, we have addressed yours’ concerns in a point by point format. 

 We look forward to hearing from you at your earliest convenience. 

Thank you for your consideration of this manuscript! 

Kind regards,

Yared Asmare Aynalem.

On behalf of authors

PONE-D-20-14158R5

Assessment of undergraduate student knowledge, attitude, and practices, towards COVID-19 in Debre Berhan University, Ethiopia

PLOS ONE

Dear Dr. Aynalem,

Thank you for submitting your manuscript to PLOS ONE. After careful consideration, we feel that it has merit but does not fully meet PLOS ONE’s publication criteria as it currently stands. Therefore, we invite you to submit a revised version of the manuscript that addresses the points raised during the review process.

Response: Dear our respected editor thanks you very much for considerations. I have submitted a revised version of the manuscript that addresses the points raised during the review process as much as we can.

The following is an odd sentence and needs to be revised in the discussion: "This might be because of easily accessible to most students at home and everywhere through the mobile internet".

ACADEMIC EDITOR COMMENT: I did not request that the sentence I questioned earlier be removed. It is important for one of the previous revisions. Please REVISE that sentence to proper English and make it understandable and resubmit this manuscript. 

Response: Dear our respected editor thanks you very much for your critical review. We have REVISE that sentence to proper English and make it understandable and resubmit this manuscript. see the color change 

 Response: Thank you. We have included ‘Response to Reviewers', 'Revised Manuscript with Track Changes ad Manuscript separately 

Response: thanks for allowing us to update the financial disclosure. However, there is no any change. 

Response: thank you for recommending depositing the laboratory protocols in protocols.io. But it is not applicable

We look forward to receiving your revised manuscript.

Kind regards,

Frank T. Spradley

Academic Editor

Response: Thank you dear editor .we has set it 

PLOS ONE

Journal Requirements:

Please review your reference list to ensure that it is complete and correct. If you have cited papers that have been retracted, please include the rationale for doing so in the manuscript text, or remove these references and replace them with relevant current references. Any changes to the reference list should be mentioned in the rebuttal letter that accompanies your revised manuscript. If you need to cite a retracted article, indicate the article’s retracted status in the References list and also include a citation and full reference for the retraction notice

Response: thank you for the feedback dear our editor.as far as we searched we didn’t cite papers that have been retracted and no any changes to the reference list. We tried to search for papers that might retract from journal. However as far as we searched, we didn’t get it. if any please suggest us we will remove and replace it .We also tried to make the reference full as the journal guide line. See the track charge 

Thank you 

Keep safe

---

## [Editor Report · Decision Letter 7]

1 Apr 2021

PONE-D-20-14158R7

Assessment of undergraduate student knowledge, attitude, and practices, towards COVID-19 in Debre Berhan University, Ethiopia

PLOS ONE

Dear Dr. Aynalem,

Thank you for submitting your manuscript to PLOS ONE. After careful consideration, we feel that it has merit but does not fully meet PLOS ONE’s publication criteria as it currently stands. Therefore, we invite you to submit a revised version of the manuscript that addresses the points raised during the review process.

SPECIFIC ACADEMIC EDITOR COMMENT:

That previous sentence in the discussion that I requested be revised is still not correct. Phrase as "This may be due to ease of access to readily updated information to most students via the internet and social media [34]."

If applicable, we recommend that you deposit your laboratory protocols in protocols.io to enhance the reproducibility of your results. Protocols.io assigns your protocol its own identifier (DOI) so that it can be cited independently in the future. For instructions see: http://journals.plos.org/plosone/s/submission-guidelines#loc-laboratory-protocols. Additionally, PLOS ONE offers an option for publishing peer-reviewed Lab Protocol articles, which describe protocols hosted on protocols.io. Read more information on sharing protocols at https://plos.org/protocols?utm_medium=editorial-emailutm_source=authorlettersutm_campaign=protocols.

We look forward to receiving your revised manuscript.

Kind regards,

Frank T. Spradley

Academic Editor

PLOS ONE
---

## [Author Response · Author response to Decision Letter 7]

2 Apr 2021

Dear Academic Editor!

Plose one 

Response to Academic Editor 

I am pleased to resubmit(R8) for publication version of “Assessment of undergraduate student knowledge, practices, and attitude towards COVID-19 in Debre Berhan University, Ethiopia for a review as original research in Plose one .The comments of the editor were highly insightful and enabled us to greatly improve the quality of our manuscript. Therefore, based on the editor’s concerns we have made extensive edition in our manuscript. The comments of the editor were highly insightful and enabled us to greatly improve the quality of our manuscript. Therefore, based on the editor’s concerns we have made extensive edition in our manuscript particularly the reference part. The formatting of the text and document (text sizes and grammatical errors) were also edited. In the following pages, we have addressed yours’ concerns in a point by point format. 

 We look forward to hearing from you at your earliest convenience. 

Thank you for your consideration of this manuscript! 

Kind regards,

Yared Asmare Aynalem.

On behalf of authors

PONE-D-20-14158R5

Assessment of undergraduate student knowledge, attitude, and practices, towards COVID-19 in Debre Berhan University, Ethiopia

PLOS ONE

Dear Dr. Aynalem,

Thank you for submitting your manuscript to PLOS ONE. After careful consideration, we feel that it has merit but does not fully meet PLOS ONE’s publication criteria as it currently stands. Therefore, we invite you to submit a revised version of the manuscript that addresses the points raised during the review process.

Response: Dear our respected editor thanks you very much for considerations. I have submitted a revised version of the manuscript that addresses the points raised during the review process as much as we can.

The following is an odd sentence and needs to be revised in the discussion: "This might be because of easily accessible to most students at home and everywhere through the mobile internet".

SPECIFIC ACADEMIC EDITOR COMMENT:

That previous sentence in the discussion that I requested be revised is still not correct. Phrase as "This may be due to ease of access to readily updated information to most students via the internet and social media [34]."

 . 

Response: Dear our respected editor thanks you very much for your critical review. We have REVISE that sentence to proper English and make it understandable and resubmit this manuscript as per your suggestion . see the color change 

 Response: Thank you. We have included ‘Response to Reviewers', 'Revised Manuscript with Track Changes ad Manuscript separately 

Response: thanks for allowing us to update the financial disclosure. However, there is no any change. 

Response: thank you for recommending depositing the laboratory protocols in protocols.io. But it is not applicable

We look forward to receiving your revised manuscript.

Kind regards,

Frank T. Spradley

Academic Editor

Response: Thank you dear editor .we has set it 

PLOS ONE

Journal Requirements:

Please review your reference list to ensure that it is complete and correct. If you have cited papers that have been retracted, please include the rationale for doing so in the manuscript text, or remove these references and replace them with relevant current references. Any changes to the reference list should be mentioned in the rebuttal letter that accompanies your revised manuscript. If you need to cite a retracted article, indicate the article’s retracted status in the References list and also include a citation and full reference for the retraction notice

Response: thank you for the feedback dear our editor.as far as we searched we didn’t cite papers that have been retracted and no any changes to the reference list. We tried to search for papers that might retract from journal. However as far as we searched, we didn’t get it. if any please suggest us we will remove and replace it .We also tried to make the reference full as the journal guide line. See the track charge 

Thank you 

Keep safe

---

## [Editor Report · Decision Letter 8]

7 Apr 2021

Assessment of undergraduate student knowledge, attitude, and practices, towards COVID-19 in Debre Berhan University, Ethiopia

PONE-D-20-14158R8

Dear Dr. Aynalem,

We’re pleased to inform you that your manuscript has been judged scientifically suitable for publication and will be formally accepted for publication once it meets all outstanding technical requirements.

Kind regards,

Frank T. Spradley

Academic Editor

PLOS ONE

---

## [Editor Report · Acceptance letter]

6 May 2021

PONE-D-20-14158R8 

Assessment of undergraduate student knowledge, attitude, and practices towards COVID-19 in Debre Berhan University, Ethiopia 

Dear Dr. Aynalem:

I'm pleased to inform you that your manuscript has been deemed suitable for publication in PLOS ONE. Congratulations! Your manuscript is now with our production department. 

Kind regards, 

on behalf of

Dr. Frank T. Spradley 

Academic Editor

PLOS ONE